# Evolution guided generative flow networks

**Zarif Ikram**
*National University of Singapore*

**Ling Pan**
*The Hong Kong University of Science and Technology*

**Dianbo Liu**
*National University of Singapore*

**Reviewed on OpenReview:** *https://openreview.net/forum?id=UgZIR6TF5N*

## Abstract

Generative Flow Networks (GFlowNets) are a family of probabilistic generative models recently invented that learn to sample compositional objects proportional to their rewards. One big challenge of GFlowNets is training them effectively when dealing with long time horizons and sparse rewards. To address this, we propose Evolution guided generative flow networks (EGFN), a simple and powerful augmentation to the GFlowNets training using Evolutionary algorithms (EA). Our method can work on top of any GFlowNets training objective, by training a set of agent parameters using EA, storing the resulting trajectories in the prioritized replay buffer, and training the GFlowNets agent using the stored trajectories. We present a thorough investigation over a wide range of toy and real-world benchmark tasks showing the effectiveness of our method in handling long trajectories and sparse rewards. We release the code at http://github.com/zarifikram/egfn.

## 1 Introduction

> "Do I contradict myself?
> Very well then I contradict myself,
> (I am large, I contain multitudes.)"

> — *Walt Whitman,* Song of Myself *(1855)*

Generative Flow Networks (GFlowNets) (Bengio et al., 2021; 2023) are a family of probabilistic amortized samplers that learn to sample from a space proportionally to some reward function $R(x)$, effectively sampling compositional objects over some probability distribution. As a generative process, it composes objects by some sequence of actions, terminating by reaching a termination state.

GFlowNets have shown great potential for diverse challenging applications, such as molecule discovery (Jain et al., 2023a), biological sequence design (Jain et al., 2022), combinatorial optimization (Zhang et al., 2023), latent variable sampling (Liu et al., 2023) and road generation (Ikram et al., 2023). The key advantage of GFlowNets over other methods such as reinforcement learning (RL) is that GFlowNets's key objective is not reward maximization, allowing them to sample diverse samples proportionally to the reward function. Although entropy-regularized RL also encourages randomness when taking actions, it is not general in when the underlying graph is not a tree (i.e., a state can have multiple parent states) (Zhao et al., 2019).

Despite the recent advancements, the real-world adaptation of GFlowNets is still limited by a major problem: temporal credit assignment for long trajectories and sparse rewards. For example, real-world problems such as protein design often are often long-horizon problems, necessitating long trajectories for sampling. Since reward is given only when the agent reaches the terminal states, associating actions with rewards over a

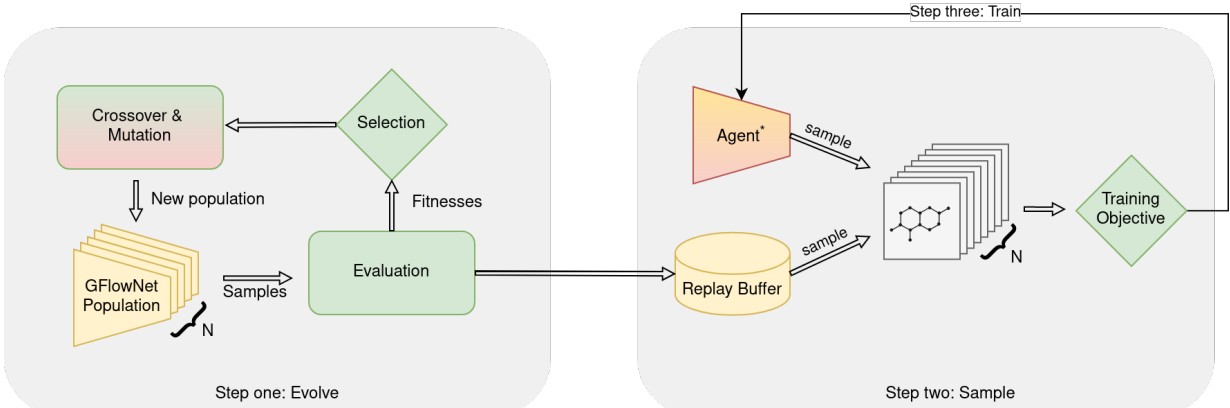

**Figure 1:** The proposed EGFN architecture. **Step one** provides high-quality trajectories to the replay buffer by evolving the agents using the trajectory rewards as fitness. **Step two** gathers training trajectories from both online and offline trajectories. **Step three** trains the star agent using the training trajectories.

lengthy trajectory becomes challenging. Additionally, reward space is sparse in real-world tasks, making temporal credit assignment more difficult. Trajectory balance (TB) objective (Malkin et al., 2022) attempts to tackle the problem by matching the flow across the entire trajectory, but in practice, it induces larger variance and is highly sensitive to sparse rewards (Madan et al., 2023), making the training unstable.

Evolutionary algorithms (EA) (Bäck & Schwefel, 1993), a class of optimization algorithms inspired by natural selection and evolution, can be a promising candidate for tackling the said challenges. Indeed, the shortcomings of GFlowNets are the advantages of EA, which makes it promising to consider incorporating EA into the learning paradigm of GFlowNets to leverage the best from both worlds. First, the selection operation in EA is achieved by fitness evaluation throughout the entire trajectory, which makes them robust to long trajectories and sparse rewards as they naturally bias towards regions with high expected returns. Secondly, mutation makes EA naturally exploratory, which is crucial for GFlowNets training and mode-finding as they rely on diverse samples for better training (Pan et al., 2022). Third, EA's natural selection biases towards parameters that generate high reward samples, which, coupled with a replay buffer, can provide sample redundancy, resulting in a better gradient signal for stable GFlowNets training.

In this work, we introduce Evolution guided generative flow networks (EGFN), a novel training method for GFlowNets combining gradient-based and gradient-free approaches and benefit from the best of both worlds. Our proposed approach is a three-step training process, as summarized in Figure 1. First, using a fitness metric across sampled trajectories taken over a population of GFlowNets agents, we perform selection, crossover, and mutation on neural network parameters of GFlowNets agents to generate a new population. To reuse the population's experience, we store the evaluated trajectories in the prioritized replay buffer (PRB). For the second step, we sample the stored trajectories from a PRB and combine them with online samples from a different GFlowNets agent. Finally, using the gathered samples, we train a GFlowNets agent using gradient descent over some objectives such as Flow matching (FM), Detailed balance (DB), and Trajectory balance (TB), where they optimize at the transition-level (FM, DB), and trajectory level (TB). The reward-maximizing capability of EA enhances gradient signal through high reward training samples, ensuring stable GFlowNets training even in conditions with sparse rewards and long trajectories. Through extensive evaluation in experimental and a wide range of real-world settings, our method proves effective in addressing weaknesses related to temporal credit assignment in sparse rewards and long trajectories, surpassing GFlowNets baselines in terms of both the number of re-discovered modes and top-K rewards. The key contributions of the work are the following.

- We effectively use EA to train GFlowNets in long trajectories and sparse rewards;

- We understand several design decisions through thorough ablation experiments to help practitioners;

- We perform extensive experiments to validate our method's capability.

## 2 Preliminaries

### 2.1 Generative Flow Networks (GFlowNets)

Generative Flow Networks (GFlowNets) are a family of generative models that samples compositional objects through a sequence of actions. We define the flow network as a single source initial state $s_0$ with in-flow $Z$, and one sink for each terminal state $x \in \mathcal{X}$ with out-flow $R(x) > 0$. We denote the Markovian composition trajectory $(s_0 \to s_1 \to \cdots \to x)$ as $\tau \in \mathcal{T}$, where $\mathcal{T}$ is the set of all trajectories. Thus, the problem is formulated as a directed acyclic graph (DAG), $(\mathcal{S}, \mathcal{E})$, where each node in $\mathcal{S}$ denotes a state with an initial state $s_0$, and each edge in $\mathcal{E}$ denotes a transition $s_t \to s_{t+1}$ with a special terminal action indicating $s = x \in \mathcal{X}$. We specifically consider DAGs that are not tree-structured, thus there exist different paths leading to the same state in the DAG, except for the root, which has no parent. Given a terminal state space $\mathcal{X}$, GFlowNets aim to learn a stochastic policy $\pi$ that can sample terminal states $x \in \mathcal{X}$ proportionally to a non-negative reward function $R(x)$, i.e., $\pi(x) \propto R(x)$. GFlowNets construct objects $x \in \mathcal{X}$ by sampling constructive, irreversible actions $a \in \mathcal{A}$ that transition $s_t$ to $s_{t+1}$. An important advantage of GFlowNets is that GFlowNets's probability of generating an object is proportional to a given positive reward for that object and we can train it in both online and offline settings, allowing us to train from replay buffers.

The key objective of GFlowNets training is to train $P_F$ such that $\pi(x) \propto R(x)$, where,

$$\pi(x) = \sum_{\substack{\tau \in \mathcal{T} \\ x \in \tau}} \prod_{t=0}^{|\tau|-1} P_F(s_{t+1}|s_t; \theta) \tag{1}$$

where, $P_F$ is a parametric model representing the forward transition probability of $s_t$ to $s_{t+1}$ with parameter $\theta$. There are several widely used loss functions to optimize GFlowNets including FM, DB and TB.

**Flow matching.** Following Bengio et al. (2021), we define the *state flow* and *edge flow* functions $F(s) = \sum_{s \in \tau} F(\tau)$ and $F(s \to s') = \sum_{\tau = (\ldots s \to s' \ldots)} F(\tau)$, respectively, where $F(\tau)$ the *trajectory flow* is a nonnegative function $F : \mathcal{T} \to \mathbb{R}^+$ so that probability measure of a trajectory $\tau \in \mathcal{T}$ is $P(\tau) = F(\tau)/\sum_{\tau \in \mathcal{T}} F(\tau)$. Then, the FM criterion matches the in-flow and the out-flow for all states $s \in \mathcal{S}$, formally –

$$\sum_{s' \in \mathrm{Parent}(s)} F(s' \to s) = \sum_{s'' \in \mathrm{Child}(s)} F(s \to s''). \tag{2}$$

To achieve the criterion, using an estimated *edge flow* $F_\theta : \mathcal{E} \to \mathbb{R}^+$, we turn equation 2 to a loss function –

$$\mathcal{L}_{\mathrm{FM}}(s; \theta) = \left[ \log \frac{\sum_{s' \in \mathrm{Parent}(s)} F_\theta(s' \to s)}{\sum_{s'' \in \mathrm{Child}(s)} F_\theta(s \to s'')} \right]^2 \tag{3}$$

**Detailed balance.** Following Bengio et al. (2023), we parameterize $F(s)$, $F(s \to s')$, and $F(s' \to s)$ with $F_\theta(s)$, $P_F(s'|s, \theta)$, and $P_B(s|s', \theta)$, respectively, where $P_F(s'|s, \theta) \propto F(s \to s')$ and $P_B(s|s', \theta) \propto F(s' \to s)$. Then, the DB loss for all $(s \to s')$ in a sampled trajectory $\tau \in \mathcal{T}$ is –

$$\mathcal{L}_{\mathrm{DB}}(s, s'; \theta) = \left[ \log \frac{F_\theta(s) P_F(s'|s, \theta)}{F_\theta(s') P_B(s|s', \theta)} \right]^2. \tag{4}$$

**Trajectory balance.** Malkin et al. (2022) extends the detail balance objective to the trajectory level, via a telescoping operation of Eq. (4). Specifically, $Z_\theta$ is a learnable parameter that represents the total flow: $\sum_{x \in \mathcal{X}} R(x) = \sum_{s:s_0 \to s \in \tau \forall \tau \in \mathcal{T}} P_F(s|s_0; \theta)$, and the TB loss is defined as:

$$\mathcal{L}_{\mathrm{TB}}(\tau; \theta) = \left[ \log \frac{Z_\theta \prod_{t=0}^{|\tau|-1} P_F(s_{t+1}|s_t, \theta)}{R(x) \prod_{t=0}^{|\tau|-1} P_B(s_t|s_{t+1}, \theta)} \right]^2. \tag{5}$$

This can incur larger variance as demonstrated in Madan et al. (2023).

We train the GFlowNets parameter $\theta$ by minimizing the loss $\mathcal{L}$ by performing stochastic gradient descent.

## 2.2 Evolutionary algorithms (EA)

Evolutionary algorithms (EA) (Bäck, 2006; Spears et al., 1993) are a class of optimization algorithms that generally rely on three key techniques: mutation, crossover, and selection as in biological evolution. The crossover operation is responsible for generating new samples based on exchange of information among a population of samples. The mutation operation alters the generated samples, usually with some probability $p_{mutation}$. Finally, the selection operation evaluates the *fitness score* of the population and is responsible for generating the next population. In this work, we apply EA in the context of the weights of the neural networks, often referred to as neuroevolution (Stanley & Miikkulainen, 2002b; Risi & Togelius, 2014; Floreano et al., 2008; Lüders et al., 2017).

## 3 Evolution guided generative flow networks (EGFN)

---

**Algorithm 1** Evolution Guided GFlowNet Training

---

**Input:**
$P_F^*$: Forward flow of the star agent with weights $\theta^*$
$pop_F$: Population of k agents with randomly initiated weights
$\mathcal{D}$ : Prioritized replay buffer
$\mathcal{E}$: Number of episodes in an evaluation
$\epsilon$: percent of greedily selected elites
$\delta$: online-to-offline sample ratio
$\gamma$: mutation strength
**for** *each episodes* **do**
  **for** *each $P_F \in pop_F$* **do**
    └ fitness, $\mathcal{D}$ = EVALUATE($P_F$, $\mathcal{E}$, *noise = None*, $\mathcal{D}$);      `// store experience in replay buffer`
  Sort $pop_F$ based on fitness in a descending order
  Select the first $\epsilon$k $P_F$ from $pop_F$ as *elite*
  Select (1 - $\epsilon$) $P_F$ from $pop_F$ stochastically based on fitness as $S$
  **while** $|S| < k$ **do**
    └ crossover between $P_F \in elite$ and $P_F \in S$ and append to $S$
  **for** *each $P_F \in S$* **do**
    └ Apply mutation $\sim \mathcal{N}(0, \gamma)$ to $\theta_{P_F}$ with probability $p_{mutation}$
  Sample a minibatch of $\delta T$ online trajectories $\mathcal{T}_{online}$ from $P_F^*$ and store them to $\mathcal{D}$
  Sample a minibatch of $(1 - \delta)T$ offline trajectories $\mathcal{T}_{offline}$ from $\mathcal{D}$
  Compute loss $\mathcal{L}$ using trajectory balance loss from $\mathcal{T}_{online} \cup \mathcal{T}_{offline}$
  Update parameters $\theta^*$ using stochastic descent on loss $\mathcal{L}$

---

EGFN is a strategy to augment existing training methods of GFlowNets. The evolutionary part in EGFN **(Step one)** samples discrete objects, e.g., a molecular structure, using a population of GFlowNets agents, evaluates the fitness of the agents based on the samples, and generates better samples by manipulating the weights of the agent population. We store the samples obtained from the population in a PRB that the GFlowNets sampler uses to, alongside on-policy samples, train its weights **(step two & three)**. To differentiate the GFlowNets agent trained by gradient descent in **step two and three** from the agent population trained using EA in **step one**, we refer to the agent trained by gradient descent as the *star agent* and GFlowNets agents trained using EA as *EA GFlowNets agents*. Algorithm 1 details the training loop, which can be summarized in the following three steps:

**Step One**   Generate a population of EA GFlowNets agents. Evaluate the fitness of the agents' weights by evaluating the samples gathered from the agents'. Apply the necessary selection, mutation, and crossover to the weights to generate the next population. Store the generated trajectories $\{(\tau_1, \ldots, \tau_{\mathcal{E}})_1, (\tau_1, \ldots, \tau_{\mathcal{E}})_2, \ldots, (\tau_1, \ldots, \tau_{\mathcal{E}})_k\}$ to the PRB.

**Step Two**   Gather online trajectories from star agent $P_F^*$ and offline samples from PRB.

**Step Three**    Train $P_F^*$ using $\{\tau_1, \tau_2, \ldots, \tau_T\}$ using gradient descent on any GFlowNets loss function such as equations 3, 4, or 5.

## 3.1   Step one: Evolve

This step involves optimizing EA GFlowNets agent weights to produce trajectories that accelerate $P_F^*$ training using the PRB. To this end, before the train begins, we initialize *pop*, a population of $k$ EA GFlowNets agents with random weights. We optimize the population weights in a standard EA process that contains selection, crossover, and mutation. Algorithm 2 in the appendix C details the evaluation process.

**Selection.**   The selection process begins with an evaluation of the population by calculating each agent's fitness scores. We define the fitness score of an agent by the mean reward of $\mathcal{E}$ trajectories $\{\tau_1, \tau_2, \ldots, \tau_{\mathcal{E}}\}$ sampled from the agent. Next, based on fitness scores, we transfer the top $\epsilon\%$ *elite* agents' weights to the next population, unmodified. Notably, we store the $k\mathcal{E}$ trajectories sampled from this step to the PRB in this step. Optionally, we restore the star agent parameters to the agent population by replacing the parameters with the worst fitness periodically.

**Crossover.**   The crossover step ensures weight mixing between agents' weights, ensuring stochasticity

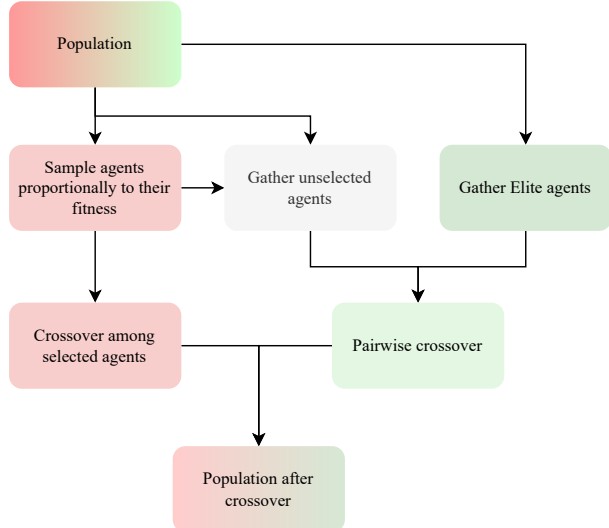

**Figure 2:** The crossover operation in EGFN. Here, we fill a proportion of population through the crossover between agents selected proportionally to their fitness. We fill the rest with the crossover between the unselected agents and the *elite* agents.

(Spears, 1995). Here, we perform the crossover in two steps. First, we perform a selection tournament process among the agents to get *pop - elite* agents, sampling proportionally to their fitness value and performing crossover among them. Next, we perform a crossover between the unselected agents and *elite*. We combine the two sets of agents and pass them on to the mutation process. To perform the crossover, we simply swap randomly chosen the weights uniformly (see details in Algorithm 3).

**Mutation.**   The mutation process ensures natural exploratory policy in agents. We apply mutation by adding a gaussian perturbation $\mathcal{N}(0, \gamma)$ to the agent weights. In this work, we only apply mutation to the non-*elite* agents.

## 3.2   Step two: Sample

In this step, we gather trajectory samples $\{\tau_1, \tau_2, \ldots, \tau_T\}$ to train the star agent. We use both online trajectories sampled by the star agent and offline trajectories stored in the PRB. For online trajectories, we construct a trajectory $\tau$ by applying $P_F^*$ to get $s_0 \to s_1 \to \cdots \to x, x \in \mathcal{X}$, where $\mathcal{X}$ is the set of all terminal states. It is noteworthy that there are many works (Rector-Brooks et al., 2023; Kim et al., 2023; Pan et al., 2023a; 2022) that augment or perturb the online trajectories by applying stochastic exploration, temperature scaling, etc. In this work, we choose a simple on-policy sampling from $P_F^*$ to get the online trajectories. For offline samples, we simply use PRB to sample trajectories collected from **step one** proportionally to the terminal reward. For this work, we take a simple approach for PRB, uniformly sampling 50% trajectories from the 20 percentile and 50% trajectories from the rest.

## 3.3   Step three: Train

We train the star agent by calculating loss $\mathcal{L}$ using equation 3, 4, or 5 and minimizing the loss by applying stochastic gradient descent to the parameter $\theta$.

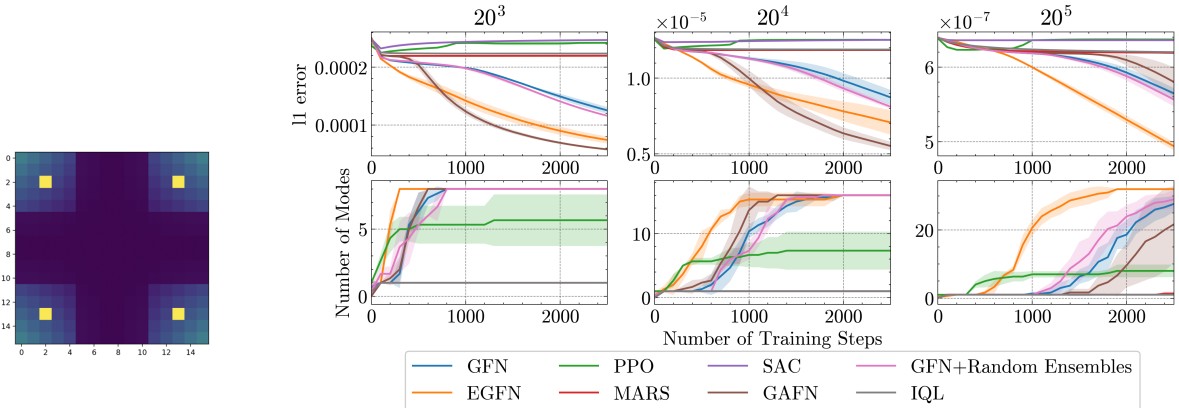

**Figure 3:** *Left:* An example hypergrid environment for dimension $D = 2$, horizon $H = 16$. Here, the $2^2 = 4$ yellow tiles refer to the high reward modes. *Right:* Experimental results comparison for the hypergrid task between EGFN, GFlowNets, RL, and MCMC baseline across increasing dimensions for 2500 training steps. *Right top:* the $\ell_1$ error between the learned distribution density and the true target density. *Right bottom:* the number of discovered modes across the training process. As the dimension of the grid increases, the trajectory length also increases. The proposed EGFN method achieves better performance than all baselines, with broader performance gap between EGFN and GFlowNets with increasing trajectory length.

## 4 Experiments

In this section, we validate EGFN for different synthetic and real-world tasks. Section 4.1 presents an investigation of EGFN's performance in long trajectory and sparse rewards, generalizability across multiple GFlowNets objectives, and an ablation study on different components. Next, we present five real-world molecule generation experiments. In section 4.2 and 4.4, we offer large-scale molecule experiments to confirm EGFN's ability to produce longer sequences. Additionally, we compare our method with experiments relevant to previous literature in section 4.3, D.2, and D.3. For all the following experiments, we use $k = 5$, $\mathcal{E} = 4$, $\epsilon = 0.2$, and $\gamma = 1$. All baselines are equipped with replay buffers that have parameters similar to the EGFN in order to conduct fair comparisons. The exception to this is PPO, where we double the sample size to guarantee fairness. All result figures report the mean and variance over three random seeds. Appendix E reports additional experiment details.

### 4.1 Synthetic tasks

We first study the effectiveness of EGFN investigating the well-studied hypergrid task introduced by Bengio et al. (2021). The hypergrid is a $D$-dimensional environment of $H$ horizons, with a $H^D$ state-space, $D + 1$ action-space, and $2^D$ modes. The $i$th action in the action space corresponds to moving 1 unit in the $i$th dimension, with the $D$th action being a termination action with which the agent completes the trajectory and gets a reward specified by -

$$R(\mathbf{x}) = R_0 + R_1 \prod_{d=1}^{D} \mathbb{I}\left[\left|\frac{\mathbf{x}_d}{H-1} - 0.5\right| \in (0.25, 0.5]\right] + R_2 \prod_{d=1}^{D} \mathbb{I}\left[\left|\frac{\mathbf{x}_d}{H-1} - 0.5\right| \in (0.3, 0.4]\right] \quad (6)$$

where $\mathbb{I}$ is the indicator function and $R_0, R_1$, and $R_2$ are reward control parameters.

In this empirical experiment, two questions interest us.

- *Does EGFN augmentation provide improvement against the best GFlowNets baseline for longer trajectories and sparse rewards?*

- *Is this method generally applicable to other baselines?*

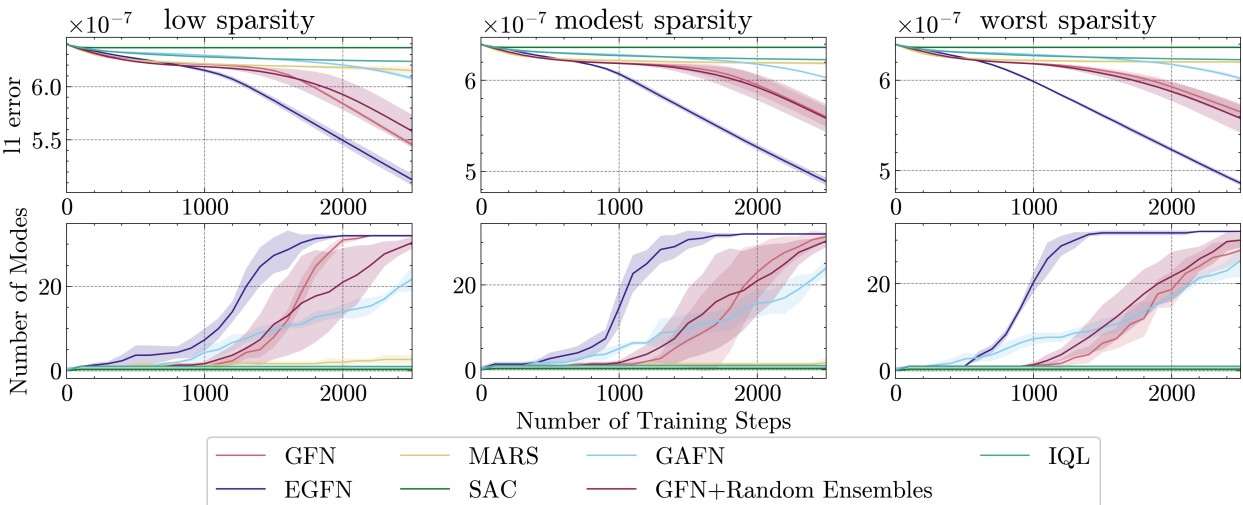

**Figure 4:** Experimental results comparison for the hypergrid task between EGFN, GFlowNets (GFlowNets, GAFN), RL (PPO, SAC, IQL), and MCMC (MARS) baseline across increasing dimensions for 2500 training steps. *Top:* the $\ell_1$ error between the learned distribution density and the true target density. *Bottom:* the number of discovered modes across the training process. Here, $H = 20, D = 5$. As the $R_0$ value decreases, the reward sparsity increases. The proposed EGFN method achieves better performance than all baselines, with broader performance gap between EGFN and GFlowNets with increasing sparsity.

**Setup**. We run all hypergrid experiments for $D \in \{3, 4, 5\}$, $H = 20$, and $R_0 \in \{10^{-3}, 10^{-4}, 10^{-5}\}$. To determine the best GFlowNets baseline, we run three objectives $\in \{\text{FM}, \text{TB}, \text{DB}\}$ (please see Figure 10 in appendix D.1) and decide to use DB with a PRB of size 1000. For a fair comparison, we use DB for implementing EGFN. We use $R_0 = 10^{-5}$ for the long trajectory experiment and $D = 5$ for the reward sparsity experiment, keeping other variables fixed. Finally, we present an ablation study on different components used in our experiments for $H = 20, D = 5$, and $R_0 = 10^{-5}$.

For a complete picture, we compare our method with RL baselines such as PPO (Schulman et al., 2017), SAC (Haarnoja et al., 2018; Christodoulou, 2019), and IQL (Kostrikov et al., 2022) and MCMC baseline such as MARS (Xie et al., 2020), and recent GFlowNet baseline such as GAFN. Besides, we also compare it against a simpler variation to our method that involves GFlowNets with its offline trajectories sampled by random policy ensembles. We provide an additional overview of the baselines in Appendix G.

**Long time horizon result.** In this experiment, as $D$ increases, $|\tau|$ increases, showing the performance over increasing $|\tau|$. Figure 3 demonstrates that EGFN outperforms GFlowNets baseline both in terms of mode finding efficiency and L1 error. Notably, as $|\tau|$ increases, the performance gap increases, confirming its efficacy in challenging environments. Unexpectedly, MARS prove to be very slow for these challenging environments. Besides, while RL baseline such PPO competes with GFlowNets and EGFN in the beginning, it fails to discover all modes due to its mode maximization objective.

**Reward sparsity result.** Next, to understand the effect of sparse rewards, we compare our method against GFlowNets for $R_0 \in \{10^{-3}, 10^{-4}, 10^{-5}\}$. With a decreasing $R_0$, reward sparsity increases. Figure 4 shows that EGFN outperforms GFlowNets. Similar to the previous experiment, we see an increasing performance gap as the reward sparsity increases. Similar to previous experiment, both RL and MCMC baselines are no match for such difficult environments.

**Ablation study result.** To understand the individual effect of each component of our method, we run the hypergrid experiment by comparing our method against the same without PRB and mutation. Figure 5 details the results of the experiment, underscoring the importance of the mutation operator in EGFN. It shows that PRB individually is not effective for improved results, but when it is coupled with mutation, our method delivers better results. Appendix F presents a detailed ablation on different parts of EA used in EGFN, showing larger population (appendix F.1), lower elite population (appendix F.2), larger replay buffer

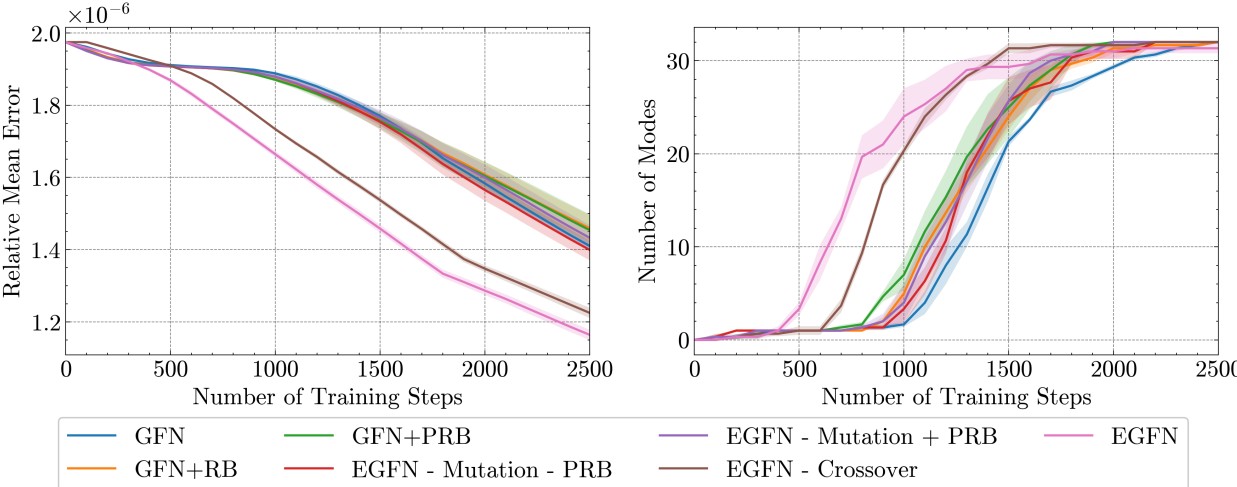

**Figure 5:** Experimental results for the hypergrid task on different components of EGFN. *Top:* the $\ell_1$ error between the learned distribution density and the true target density. *Bottom:* the number of discovered modes across the training process. Here, $H = 20, D = 5$, and we use DB objective.

(appendix F.3), greater mutation strength (appendix F.4), higher priority percentile (appendix F.5), and high priority-non-priority split (appendix F.6) generally improves EGFN.

## 4.2 Antibody sequence optimization task

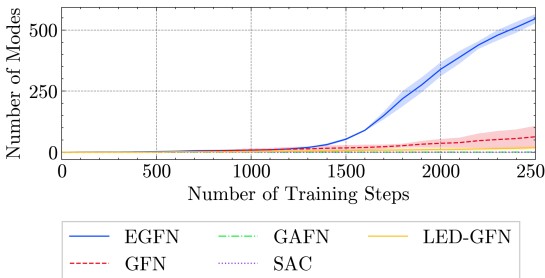

**Figure 6:** Experimental results comparison for antibody sequence optimization task for 2500 training steps. EGFN achieves the best performance in terms of mode discovery.

|  | Top-K $E_d$ ↑ | Top-K *index* ↓ |
|---|---|---|
| GFlowNet | 34.91 | 39.04 |
| EGFN | **40.47** | **35.95** |
| GAFN | 17.46 | 38.08 |
| SAC | 0.0 | 40.41 |

**Table 1:** Experiment results for antibody sequence optimization between EGFN, GFlowNets, GAFN, and SAC baseline task after training for 2500 steps. EGFN achieves the lowest instability index while retaining the best diversity. The performances measures are taken from 1000 samples after training and $K = 100$

**Setup.** In this task, our goal is to generate an antibody heavy chain sequence of length 50 that, augmented with a pre-defined suffix sequence, optimizes the instability index (Guruprasad et al., 1990) of the antibody sequence. We represent antibody sequences as $x = (x_1, \ldots, x_d)$, where $x_i \in 1, \ldots, 21$ refers to the 20 amino acid (AA) type, with the gap token '-' to ensure variable length. We label the training sequences using BioPython (Cock et al., 2009) and transform the instability index to a reward by performing the following transformation: $R(x) = 2^{\frac{index-5}{10}}$ where *index* is the instability index of $x$. Finally, we define sequences with an instability index less than 35 as modes[1].

**Results.** For comparison, we include GFlowNets, GAFN, and SAC. We train all the baselines for 2500 steps and sample 1000 sequences afterwards. To calculate the diversity of the samples, we use the edit distance

---

[1]Based on the literature, any index greater than 40 is unstable, but we define modes with *index* < 35 to make the reward space more sparse.

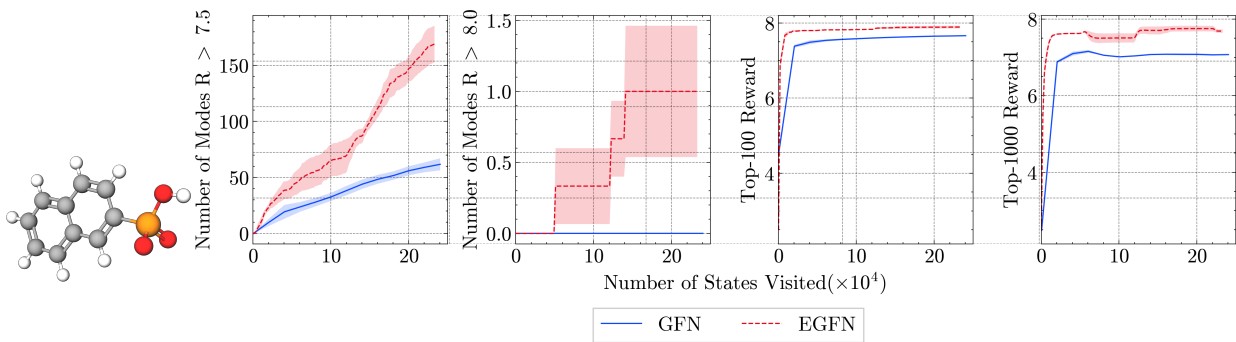

**Figure 7:** From left to right. Example binder produced in the Soluable Epoxy Hydrolase (sEH) binder generation task. Here the structure corresponds to the molecule with SMILES representation `O=P([O-])(O)c1ccc2ccccc2c1`. sEH binder generation experiment over $2.5 \times 10^4$ training steps. GFlowNets implementation uses FM objective. The number of modes with a reward threshold of 7.5 and 8.0. The average reward across the top-100 and 1000 molecules. The proposed augmentation with EGFN achieves better results both in terms of mode discovery and average reward.

($E_d$). Figure 4.2 reports the results from both training and post-training. During training, EGFN discovers a significant number of modes compared to the other baselines. We can see that this translates to post-training sample quality, as EGFN samples have a lower mean instability index and higher diversity.

### 4.3 Soluable Epoxy Hydrolase (sEH) binder generation task

**Setup.** In this experiment, we are interested in generating molecules with desired chemical properties that are not too similar to one another. Here, we represent molecules states as graph structures and actions as a vocabulary of blocks specified by junction tree modeling (Bickerton et al., 2012; Shi et al., 2020). In the pharmaceutical industry, drug-likeliness (Bickerton et al., 2012), synthesizability (Shi et al., 2020), and toxicity are crucial properties. Hence, we are interested in finding diverse candidate molecules for a given criteria to increase chances for post-selection. Here, the criteria is the molecule's binding energy to the 4JNC inhibitor of the soluble epoxide hydrolase (sEH) protein. To this end, we train a proxy reward function for predicting the negative binding energy that serves as the reward function. We perform the experiment following the experimental details and reward function specifications from Bengio et al. (2021). Since we are interested in both the diversity and efficacy of drugs, we define a mode as a molecule with a reward greater than 7.5 and a tanimoto similarity among previous modes less than 0.7. We use FM as the GFlowNets baseline and implement EGFN for the same objective.

**Results.** Since the state space is large, we show the result of the number of modes $> 7.5$, the number of modes $> 8.0$, the top-100, and the top 1000 over the first $2.5 \times 10^4$ states visited. Figure 7 confirms that EGFN outperforms GFlowNets baseline for mode discovery. Remarkably, EGFN discovers rare molecule with a very high reward ($R > 8$) that GFlowNets fails to discover. Besides, EGFN has a better top-100 and top-1000 reward performance than GFlowNets baseline, soliciting its mode diversity.

### 4.4 hu4D5 CDR H3 mutation generation task

**Setup.** To further show the robustness, here we consider the task of generating CDR mutants on a hu4D5 antibody mutant dataset (Mason et al., 2021). After de-duplication and removal of multi-label samples, this dataset contains 9k binding and 25k non-binding hu4D5 CDR mutants up to 10 mutations. To classify the generated samples, we train a binary classifier that achieves 85% accuracy on an IID validation set. For this task, we define samples, which are classified as binders with a high probability ($> 96\%$) as modes. The reward transformation in this task is $R(x) = 2^{p_{bind}} - 1$.

**Results.** We compare EGFN against GFlowNets, GAFN, and SAC and train the baselines for 2500 steps. As shown in Figure 8, the modes discovered by EGFN during training exceed the other baselines.

## 4.5 Result summary

In both the synthetic and real-world experiments, EGFN performs well for mode discovery using fewer training steps than GFlowNets baseline. The performance gap increases with increasing trajectory length and reward sparsity. We also discover that the mutation operator is the most important factor for performance improvement.

## 5 Discussion and Limitations

*Why does EGFN work?* To explore this, we compare the trajectories stored in the training step for GFlowNets and EGFN across training steps $\in \{500, 1000, 1500, 2000, 2500\}$ in the hypergrid task in Figure 9 for different levels of sparsity ($R_0 \in \{10^{-2}, 10^{-5}\}$. We see that when there is low reward sparsity (left), the training trajectory length distribution of both methods is similar. However, when the reward sparsity increases (right), GFlowNets training trajectories center around a lower trajectory length than that of EGFN. However, for a reward-symmetrical environment like hypergrid, the sampled trajectory length must be uniformly distributed to truly capture the data distribution. EGFN can achieve this through population variation, diversifying the sampled trajectories in the PRB.

A potential limitation of this work is that it is based on GFlowNets, which is an amortized sampler with the provable objective of fitting the reward distribution instead of RL's reward maximization. Thus, depending on the perspective, this method may seem slower than RL. Besides, because of the evolution step, the method takes longer than the vanilla GFlowNets. For example, we report the runtime analysis on the QM9 task (appendix D.3) in table 2, which is performed with an Intel Xeon Processor (Skylake, IBRS) with 512 GB of RAM and a single A100 NVIDIA GPU. We can see that the runtime has increased by 35%. However, we do note that there is no additional cost for increasing the population size, except for increased memory requirement, since we perform the population sampling by utilizing threading.

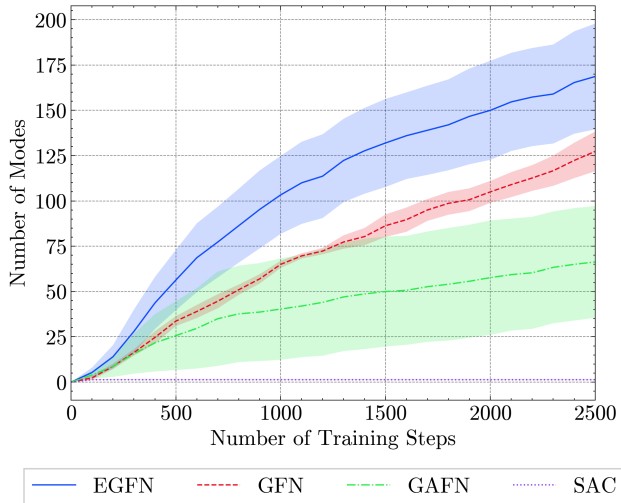

**Figure 8:** Experimental comparison of discovered modes across the training process for the hu4D5 CDR H3 mutation generation task between EGFN, GFlowNets, SAC, and GAFN baseline for 2500 training steps. *Right top:* the $\ell_1$ error between the learned distribution density and the true target density. EGFN augmentation achieves better mode discovery compared to other baselines.

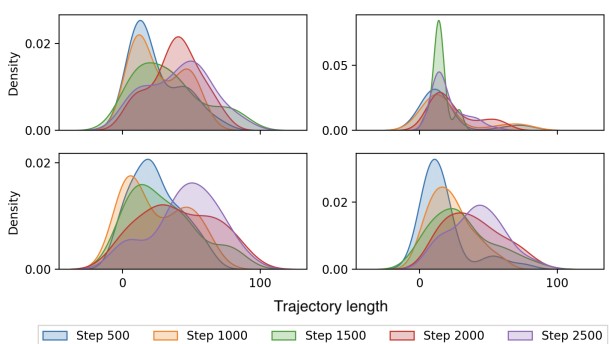

**Figure 9:** Experimental results for the trajectory lengths of the trajectories stored in the training samples across different training steps for $R_0 = 10^{-2}$ (left) and $10^{-4}$ (right). *Top:* GFlowNets *Bottom:* EGFN

Another avenue where our work falls short is that it *does not* propose a better credit assignment solution but rather utilizes the quality-diversity samples from EA to address the challenge of training EGFN in long time horizons and sparse rewards. Figure 22 motivates this approach, showing that the population achieves high mean reward quicker than the star agent.

Finally, while we keep the total training samples in each step the same across the baselines for our experiments, we do not strictly enforce the number of reward calls to be same. However, they are also roughly the same, e.g., EGFN uses 36 calls while other baselines use 32, except for PPO which doubles the reward calls since

**Table 2:** Runtime analysis comparing EGFN and GFlowNets on the QM9 task.

|      | Wall-clock time per step | K | number of eval samples | number of online samples |
|------|--------------------------|---|------------------------|--------------------------|
| EGFN | $3.033 \pm 0.004$        | 10 | 1                     | 24                       |
| GFN  | $2.243 \pm 0.016$        | 0 | 0                      | 32                       |

we double the amount of on-policy samples due to not having off-policy samples. This said, we still see significant improvement of EGFN compared of GFlowNets when we plot the results with number of trajectory evaluations on the X axis (Figure 21 in Appendix H).

## 6 Related work

### 6.1 Evolution in learning

There has been many attempts to augment learning, especially RL, with EA. Early works such Whiteson (2006) combine NEAT (Stanley & Miikkulainen, 2002a) and Q Learning (Watkins & Dayan, 1992) by using evolutionary strategies to better tune the function approximators. In a similar manner, Colas et al. (2018) uses EA for exploration in policy gradient, generating diverse samples using mutation. Fernando et al. (2017) use EA for allowing parameter reuse without catastrophic forgetting. Recently, many methods use EA to enhance deep RL architectures such as Proximal Policy Optimization (PPO) (Hämäläinen et al., 2020), Soft-Actor Critic (SAC) (Hou et al., 2020), and Policy Gradient (Khadka & Tumer, 2018). The key idea from these approaches is to use EA to overcome the temporal credit assignment and improve exploration by getting diverse samples (Lee et al., 2020), with some exceptions such as Gangwani & Peng (2018); Fujimoto et al. (2018); Pourchot & Sigaud (2019) where they utilize EA to tune the parameter of the actor itself.

### 6.2 GFlowNets

GFlowNets have recently been applied to various problems (Liu et al., 2023; Bengio et al., 2021). There have also been recent efforts in extending GFlowNets to continuous (Lahlou et al., 2023) and stochastic worlds (Pan et al., 2023b), and also leveraging the power of pre-trained models (Pan et al., 2024). In GFlowNets training, exploration is an important concept for training convergence, which many works attempt in different ways. For example, Bengio et al. (2021) use $\epsilon$-greedy exploration strategy, Kim et al. (2023) learn the logits conditioned on different annealed temperatures, Pan et al. (2022) introduces augmented flows into the flow network represented by intrinsic rewards, etc. The temporal credit assignment for long trajectories and sparse reward is a more recently studied topic for GFlowNets. Recent works such as Malkin et al. (2022) attempt to tackle this problem by minimizing the loss over an entire trajectory as opposed to state-wise FM proposed by Bengio et al. (2021), however, it may incur large variance as demonstrated in Madan et al. (2023).

## 7 Conclusion

In this work, we presented EGFN, a simple and effective EA based strategy for training GFlowNets, especially for credit assignment in long trajectories and sparse rewards. This strategy mixes the best of both worlds: EA's population-based approach biases towards regions with long-term returns, and GFlowNet's gradient-based objectives handle the matching of the reward distribution with the sample distribution by leveraging better gradient signals. Besides, EA promotes natural diversity of the explored region, removing the need to use any other exploration strategies for GFlowNets training. Furthermore, by incorporating PRB for offline samples, EA promotes redundancy of high region samples, stabilizing the GFlowNet training with better gradient signals. We validate our method on a wide range of challenging toy and real-world benchmarks with exponentially large combinatorial search spaces, showing that our method outperforms the best GFlowNet baselines on long time horizon and sparse rewards.

In this work, we implement a standard evolutionary algorithm for EGFN. Incorporating more complex sub-modules of EA for GFlowNets training such as Covariance Matrix Adaptation and Evolution Strategy (CMA-ES) such as the work in Pourchot & Sigaud (2019) can be an exciting future work. Another future direction could be integrating the gradient signal from the GFlowNets objectives into the EA strategy, creating a feedback loop. Besides, while we use a reward-maximization formulation for the EA in this work, there are works such as Parker-Holder et al. (2020) that directly improves diversity by formulation. We leave that for the future work.

### Broader Impact Statement

The primary goal of this work is to advance the field of Machine Learning. The authors, thus, do not foresee any specific negative impacts of this work.

### Acknowledgments

DL is funded by NUS-UToronto joint grant. Furthermore, ZI is thankful to his family. The authors are grateful to Emmanuel Bengio and Moksh Jain for their valuable feedback and suggestions in this work.

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

# A Codes

The hypergrid and sEH binder task is based on the code from https://github.com/zdhNarsil/Distributional-GFlowNets. The QM9 and TFBind8 task is based on the code from https://github.com/maxwshen/gflownet. All our implementation code uses the PyTorch library (Paszke et al., 2019). We used MolView https://molview.org/ to visualize the molecule diagrams for our paper.

# B Summary of notations

We summarize the notations used in our paper in the table 3 below.

**Table 3:** Notations summary

| Symbol | Description |
|--------|-------------|
| $\mathcal{S}$ | state space |
| $\mathcal{X}$ | terminal state space |
| $\mathcal{A}$ | action space $(s \rightarrow s')$ |
| $\mathcal{T}$ | trajectory space |
| $s_0$ | initial state in $\mathcal{S}$ |
| $s$ | state in $\mathcal{S}$ |
| x | terminal state in $\mathcal{X}$ |
| $\tau$ | trajectory in $\mathcal{T}$ |
| $P_F$ | forward flow |
| $P_B$ | backward flow |
| $k$ | population size |
| $\mathcal{D}$ | replay buffer |
| $\epsilon$ | elite population ratio |
| $\gamma$ | mutation strength |

# C Fitness evaluation algorithm

---
**Algorithm 2** Evaluation of Forward Flows
---
**Data:** Forward flow $P_F$
**Result:** Updated replay buffer with trajectories and fitness of $P_F$
**Procedure** EVALUATE($P_F, \mathcal{E}, \mathcal{D}$)
    fitness $\leftarrow 0$

    **for** *iter = 1 to $\mathcal{E}$* **do**
        Initialize start state $s$ ;                    // Can also be parallelized
        Initialize trajectory $\mathcal{T}$ to an empty list
        **while** *s not a terminal state* **do**
            Sample action $a$ based on $P_F(s|\theta_{P_F})$
            $s' \leftarrow transition(s, a)$
            Append $(s', a)$ to the $\mathcal{T}$
            $s \leftarrow s'$

        Compute $\mathcal{R}$(s) using reward function using the last state in $\mathcal{T}$
        fitness $\leftarrow$ fitness $+ \mathcal{R}$(s)
        Append $\mathcal{T}$ to $\mathcal{D}$

    Return $\frac{\text{fitness}}{\mathcal{E}}, \mathcal{D}$

---

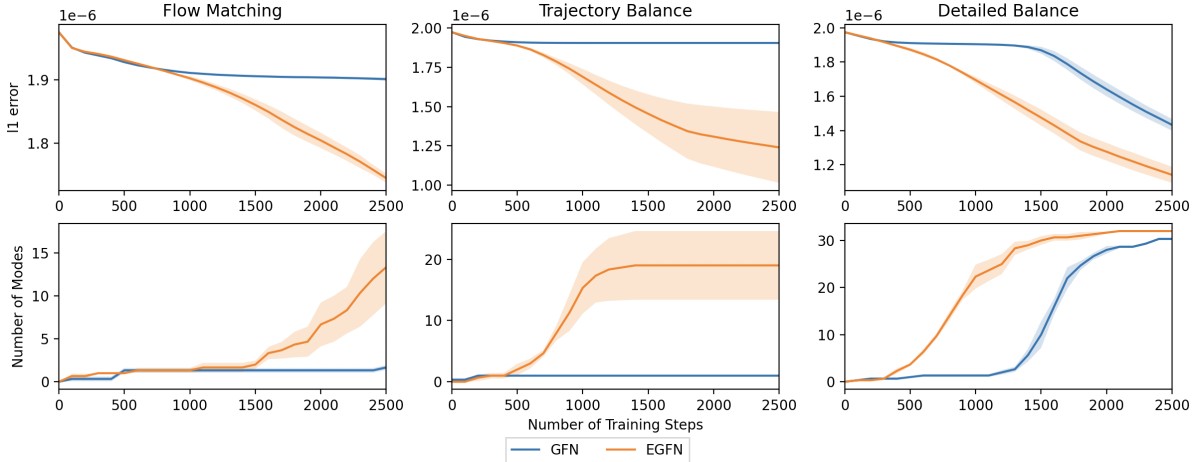

**Figure 10:** Experimental results for the hypergrid task between EGFN and GFN across different GFlowNets objectives. *Top:* the $\ell_1$ error between the learned distribution density and the true target density. *Bottom:* the number of discovered modes across the training process. The proposed augmentation with EGFN achieves better results for all three objectives.

# D    Additional Experiments

## D.1    Generalizability experiment

To see how well EGFN works with different GFlowNets objectives, we show the result of augmentation of our method over all three GFlowNets objectives in Figure 10. We see that EGFN offers a steady improvement across all three GFlowNets objectives.

## D.2    Transcription factor binder generation task

**Setup** In this experiment, we generate a nucleotide as a string of length 8. Although the string could be generated autoregressively, in this experiment setting, we use a Prepend-Append Markov decision process (PA-MDP), used in similar settings by Shen et al. (2023); Ikram et al. (2023). Using this MDP, GFlowNets agent actions prepend or append to the nucleotide string. The reward is a DNA binding affinity to a human transcription factor provided by Trabucco et al. (2022). We attempt three GFlowNets objectives, finally deciding to use TB as the best GFlowNets baseline and implement EGFN with the same using a reward exponent $\beta = 3$.

**Results** Figure 11 shows the result over 2000 training steps, showing that GFlowNets outperforms GFlowNets baseline both in terms of the number of modes discovered and the mean relative error.

## D.3    Small molecule generation Task

**Setup** In this experiment, we generate a small molecule graph based on the QM9 data (Ramakrishnan et al., 2014) that maximizes the energy gap between its HOMO and LUMO orbitals, thereby increasing

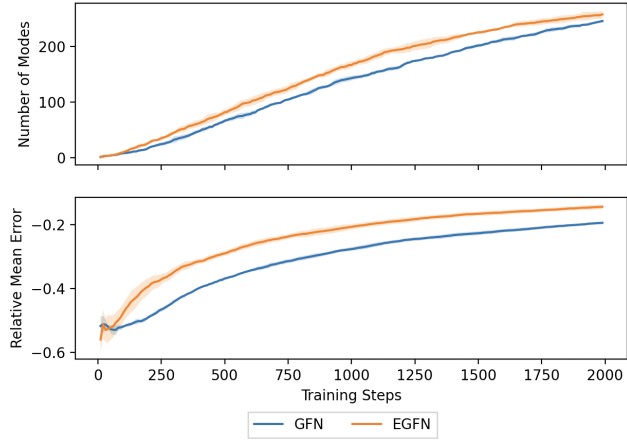

**Figure 11:** Experiment results for transcription factor binder generation task over 2000 training steps for $\beta = 3$. *Top:* the number of discovered modes across the training process. *Bottom:* the relative mean error. The proposed augmentation with EGFN achieves better results. GFlowNets implementation uses TB objective.

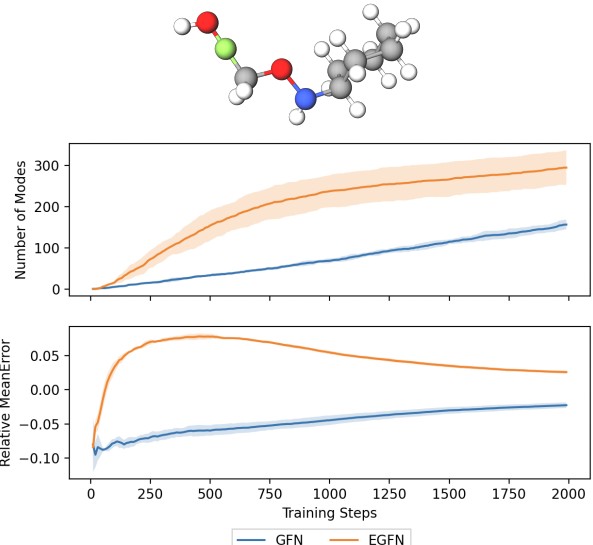

**Figure 12:** From top to bottom: Example molecule produced in the qm9 task with SMILES representation `C1CCCCC1NOCFO`. Experiment results for small molecule generation task on the QM9 data over 2000 training steps for $\beta = 1$. The number of discovered modes across the training process. The relative mean error. The proposed augmentation with EGFN achieves better results. GFlowNets implementation uses TB objective.

its stability. The resulting molecule is a 5-block molecule, having a choice among 12 blocks for its two stems. For the reward function, we use a pre-trained MXMNet proxy by Zhang et al. (2020) with a reward exponent $\beta = 1$. Similar to D.2, we use TB for this experiment.

**Results** In Figure 12, we report the mode discovery and L1 error results over 2000 training steps. Similar to previous experiments, EGFN maintains a steady improvement over the GFlowNets baseline for mode discovery while decreasing the L1 error quicker.

# E   Additional implementation details

## E.1   Hypergrid task

In our experiments, $R_1$ and $R_2$ stay at a fixed value of 0.5 and 2. In our experiments, $R_0$ varies within $\{10^{-3}, 10^{-4}, 10^{-5}\}$. A mode is the terminal state $\mathbf{x}$ for which $R(\mathbf{x}) = R_{max}$. From the equation 6, it is evident that there are $2^D$ distinct reward regions, with each region having $M$ number of modes ($M = 1$ for our experiments). Besides, $H$ refers to the *horizon* of the environment, meaning each dimension of $\mathbf{x}$ can be equal to $i \in \{0, 1, 2, \dots, H-1\}$. For example, Figure 4 uses $D = 5$ and $H = 20$. Clearly, while increasing both $D$ and $H$ increases the complexity of the task, effecting the trajectory length $|\tau|$ and the number of states $|\mathcal{X}|$, only increasing $D$ increases the number of modes. To calculate the empirical probability density, we collect the past visited 200000 states and calculate the probability density.

**Architecture** We model the forward layer with a 3-layer MLP with 256 hidden dimensions, followed by a leaky ReLU. The forward layer takes the one-hot encoding of the states as inputs and outputs action logits. For FM, we simply use the forward layer to model the edge flow. For TB and DB, we double the action space and train the MLP as both the forward and backward flow. We use a learning rate of $10^{-4}$ for FM and $10^{-3}$ for both TB and DB, including a learning rate of 0.1 for $Z_\theta$. The replay buffer uses a maximum size of 1000, and we use a *worst-reward first* policy for replay replacement. For RL and MCMC baselines, we use the implementation provided by Bengio et al. (2021). For GAFN baseline, we use the official code provided by Pan et al. (2022), performing a hyper-parameter tuning of intrinsic reward and using intrinsic reward = $\{0.05, 0.03, 0.01\}$ for our experiments with $D \in \{3, 4, 5\}$, respectively.

### E.2   Antibody sequence optimization task

In this task, we are interested in generating the mutation of the first 50 characters of a heavy-chain antibody sequence. To adapt to the variable length of the sequence, we use a gap token of '-', making the total number of actions 21 [2]. The heavy and light chain suffixes appended to the generated 21-length sequences are QVQLVQSGTEVKKPGSSVKVSCKASGGTFSSYAVSWVRQAPGQGLEWMGRFIPIL-NIKNYAQDFQGRVTITADKSTTTAYMELINLGPEDTAVYYCARGSLSGREGLPLEYWGQGTLV SVSS and EVVMTQSPATLSVSPGESATLYCRASQIVTSDLAWYQQIPGQAPRLLIFAASTRATGI-PARFSGSGSETDFTLTISSLQSEDFAIYYCQQYFHWPPTFGQGTKVEIK, respectively. We collect the chain pair from the observed antibody space (OAS) database. Finally, we use a reward policy to reduce the instability index of the mutated sequences. The main goal of this task is to assess the GFlowNet's ability to perform on longer sequences with long trajectories. To the best of our knowledge, no GFlowNets literature has experimented with tasks with such a long trajectory length, mainly because GFlowNets struggle with longer trajectories.

**Architecture** This task's architecture closely resembles the hypergrid setting. Careful architecture choice may improve the observed results of the task, but this is outside the scope of this work.

### E.3   sEH binder generation Task

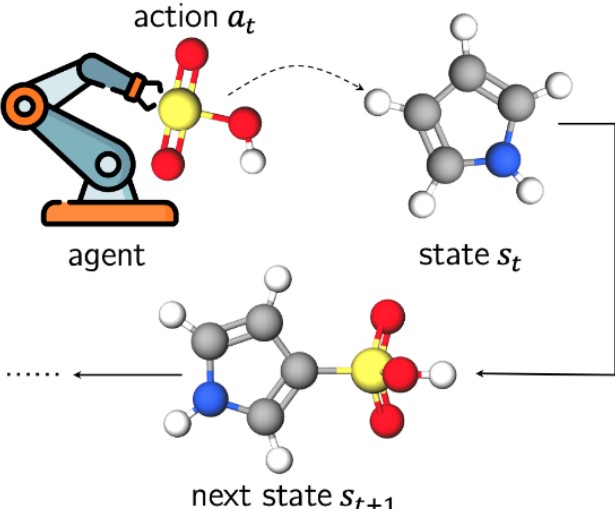

**Figure 13:** Illustration of GFlowNets policy for sEH binder generation task. Figure adopted from Pan et al. (2022)

For this task, the number of actions is within 100 to 2000, depending on the state, making $|\mathcal{X}| \approx 10^{16}$. We allow the agent to choose from a library of 72 blocks. Similar to Bengio et al. (2021), we include the different symmetry groups of a given block, making the action count 105 per stem. We also allow the agent to select up to 8 blocks, choosing them as suggested by Sterling & Irwin (2015) from the ZINC dataset (Sterling & Irwin, 2015). Following Zhang et al. (2024), we use Tanimoto similarity, defined by the ratio between the intersection and the union of two molecules based on their SMILES representation. To maintain diversity, we define a mode to be a terminal state for which the normalized negative binding energy to the 4JNC inhibitor of the soluble epoxide hydrolase (sEH) protein is more than 7.5 and the tanimoto similarity of other discovered modes is less than 0.7. Note that this objective is more limiting than simply counting the number of different Bemis-Murcko scaffolds that reach the reward threshold like Bengio et al. (2021). Since we are focusing on both molecule separation and optimization, our approach is more applicable for de novo molecule design, while the scaffold-based metric is suitable for lead optimization.

---

[2] $\mathcal{A} = 22$, as we still need a *stop token* for the GFlowNets agent.

**Architecture** Following Bengio et al. (2021), we use a message passing neural network (MPNN) (Gilmer et al., 2017) that receives the atom graph to calculate the proxy reward of the molecules. Similarly, we use another MPNN that receives the block graph for flow estimation. The block graph is a tree of learned node embeddings that represent the blocks and edge embeddings that represent the bonds. To represent the flow, we pass the stems through a 10-layer graph convolution followed by GRU to calculate their embedding and pass the embedding through a 3-layer MLP to get a 105-dimension logit. Similarly, to represent the stop action, we pass the global mean pooling to the 3-layer MLP. The MLPs use 256 hidden dimensions, followed by a leakyReLU. We use a learning rate of 0.0005 and a minibatch size of 4. For EGFN, we use an offline sample probability of 0.2. Besides, we use a reward exponent $\beta = 10$ and a normalizing constant of 8.

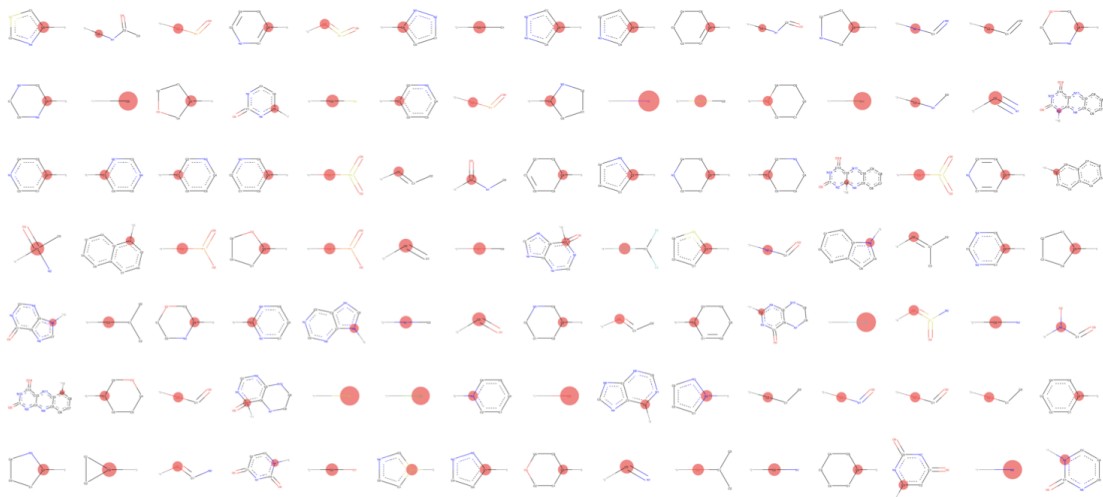

**Figure 14:** Molecule blocks for the sEH protein task. Figure adapted from Bengio et al. (2021)

### E.4 hu4D5 CDR H3 mutation generation task

Following Frey et al. (2023); Ikram et al. (2024), we consider the task of generating CDR3 10-length mutations on a hu4D5 antibody mutant dataset. The dataset contains both non-binder and binder sequences. We train a discriminator model on the binding likelihood of a sequence and then compare the starting points for making sequences that improve both the binding likelihood and the diversity shown by the pairwise edit distance of the new sequences.

**Archtecture** The architecture of different baselines follows the baselines of the previously discussed tasks. We use a 35-layer Bytenet architecture with a hidden layer of 128 for the discriminator model. This is followed by a 3-layer 1D-CNN and a 3-layer MLP, with leakyReLU activations added between the layers. We train the model on the hu4D5 antibody mutant dataset, which achieves 85% accuracy on the test set.

### E.5 TFBind8 task

For this task, the goal is to generate an 8-length DNA sequence that maximizes the binding activity score with a particular transcription factor SIX6REFR1 (Barrera et al., 2016). We use a precalculated oracle for the proxy reward calculation. Using a PA-MDP, we prepend or append a neucleotide in each step. Note that this formulation reduces the trajectory length significantly despite our effort to showcase better performance in long trajectories, but we use it following previous works.

**Architecture** Following Shen et al. (2023), the GFlowNets architecture uses a 2-layer MLP with 128-dimension hidden layer parameterizing SSR ($\mathcal{S}, \mathcal{S}' \to \mathbb{R}^+$). For each training step, we train on both online and offline trajectories for three steps, using a minibatch of 32. Besides, we use a learning rate of $10^{-4}$ for policy and 0.01 for $Z_\theta$. Finally, we use a reward exponent $\beta = 3$ and an exploration probability of 0.01 (we do not use any exploration for EGFN).

### E.6 QM9 task

The goal here is to generate diverse molecules based on the QM9 data (Ramakrishnan et al., 2014) that maximize the HOMO-LUMO. To that end, we use the reward proxy that Jain et al. (2023b) provides based on Zhang et al. (2020). Similar to the sEH task, we generate molecules with atoms and bonds. The blocks used here are the following: `C, O, N, C-F, C=O, C#C, c1ccccc1, C1CCCC1, C1CCNC1, CCC`.

**Architecture** Using a PA-MDP, we use a 2-layer MLP with 1024 hidden dimensions for flow estimation. The reward proxy is a MXMNet proxy trained on the QM9 data. We use a reward exponent $\beta = 1$. The learning rate and training style follow the ones used for the TFBind8 task, with the exception of exploration probability (0.1 here) and hidden dimension (1024 here).

We detail the summary of the training hyperparameters in table 4.

**Table 4:** Summary of the hyperparameters for all experiments

|  | Hypergrid/Antibody/CDR3 | sEH Small Molecules | TFBind8 | QM9 |
|---|---|---|---|---|
| Learning Rate | $10^{-4}$ (FM), $10^{-3}$ | $5 \times 10^{-4}$ | $10^{-4}$ | $10^{-4}$ |
| $Z_\theta$ Learning Rate | 0.1 | N/A | 0.01 | 0.01 |
| $\beta$ | 1 | 10 | 3 | 1 |
| MDP | Enumerate | Sequence Insert | PA-MDP | PA-MDP |
| Exploration $\epsilon$ (none for EGFN) | 0 | 0 | 0.01 | 0.1 |
| Replay Buffer Training | 50% | 0 (20% for egfn) | 50% | 50% |
| MLP layers | 3 | 3 | 2 | 2 |
| MLP hidden dimensions | 256 | 256 | 128 | 1024 |

## F Additional ablation experiments

### F.1 Number of population

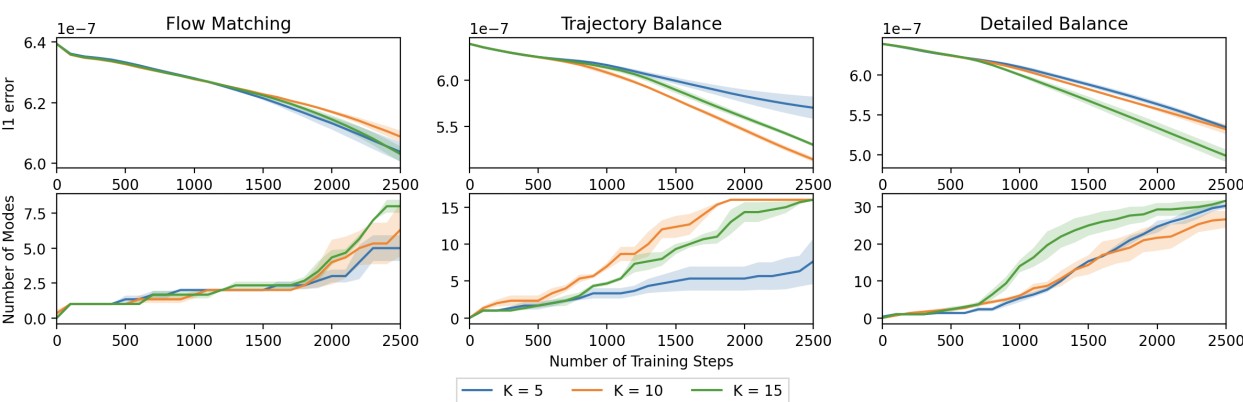

**Figure 15:** Experimental results for the hypergrid task for EGFN among different values of $k$ across the three training objectives. *Top:* the $\ell_1$ error between the learned distribution density and the true target density. *Bottom:* the number of discovered modes across the training process. Here, $H = 20, D = 5$. The results show that increasing population size leads to diminishing returns.

To investigate the effect of population size, we vary the $k \in \{5, 10, 15\}$, while keeping $\epsilon = 0.2, D = 5, H = 20, R_0 = 10^{-5}$. We plot the results of the experiment in Figure 15. It shows that increasing $k$ beyond 10 leads to diminishing returns, motivating our choice of $k = 10$ for all the experiments. For DB, however, increasing $k$ leads to considerable improvement. Indeed, this is useful because increased population size leads to more evaluation round required. While these evaluation round can be parallelized with threads as we do in our work, massive population size requirement is difficult to satisfy.

> **Takeaway:** FM and DB benefit from higher sample diversity provided by larger population.

## F.2 Elite population

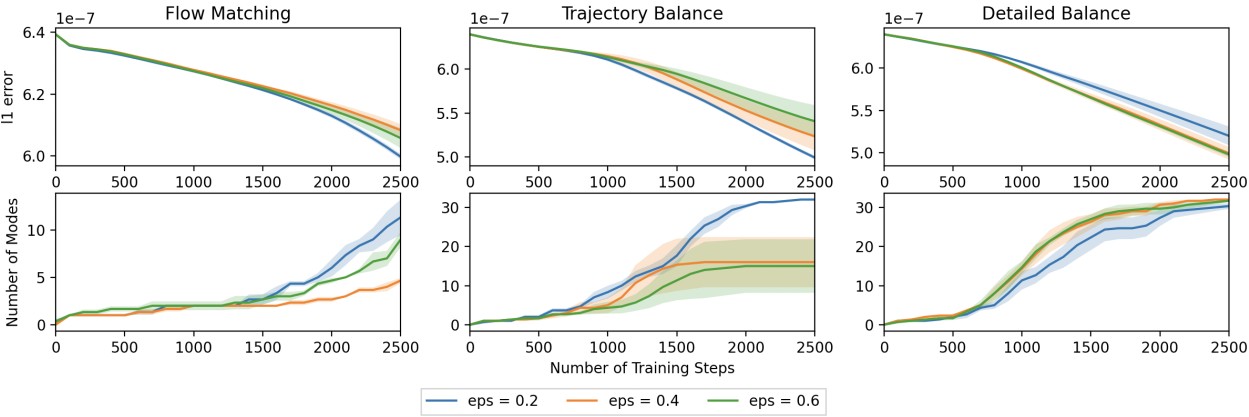

**Figure 16:** Experimental results for the hypergrid task for EGFN among different values of $\epsilon$ across the three training objectives. *Top:* the $\ell_1$ error between the learned distribution density and the true target density. *Bottom:* the number of discovered modes across the training process. Here, $H = 20, D = 5$. We observe that lower $\epsilon$ leads to better results for FM and TB.

Following ablation on $k$, we next perform ablation on the elite population ratio, $\epsilon \in \{0.2, 0.4, 0.6\}$. For this experiment, we use the same hypergrid settings for $k = 10$. We plot the results of the experiment in Figure 16. It shows that low $\epsilon$ improves mode discovery, especially for FM and TB objectives. This result is reasonable: we apply mutation and crossover only to the non-elite population, so having a low number of elite population means we have a better chance at exploring using the non-elite population's mutation and crossover.

> **Takeaway:** FM and TB benefit from lower mode-seeking behavior through lower number of elite population.

## F.3 Replay buffer size

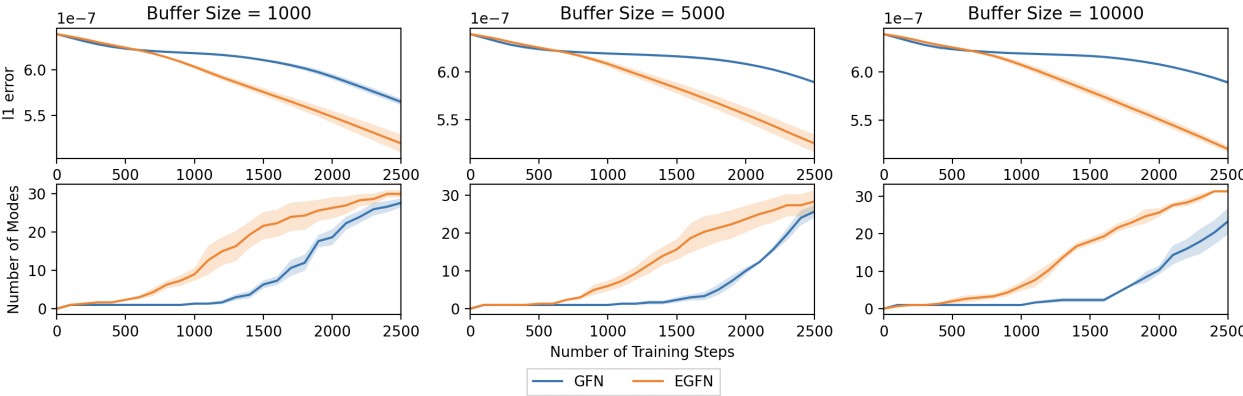

**Figure 17:** Experimental results for the hypergrid task between GFlowNets and EGFN across different values of $|\mathcal{D}|$. *Top:* the $\ell_1$ error between the learned distribution density and the true target density. *Bottom:* the number of discovered modes across the training process. Here, we use DB for $H = 20, D = 5$. The results show that while increasing replay buffer size improves the robustness of the result, it has little effect otherwise.

To understand the effect of replay buffer size, we run GFlowNets baseline with PRB and EGFN on a 20x20x20x20x20 environment with $R_0 = 10^{-5}$ for replay buffer size $\in \{1000, 5000, 10000\}$. We present

the findings in Figure 17. From the figure, we see that increasing buffer size generally has little effect for GFlowNets, but it improves EGFN's robustness a little.

> **Takeaway:** Higher replay buffer capacity preserves more diverse samples, resulting in a more consistent EGFN result.

## F.4 Mutation strength

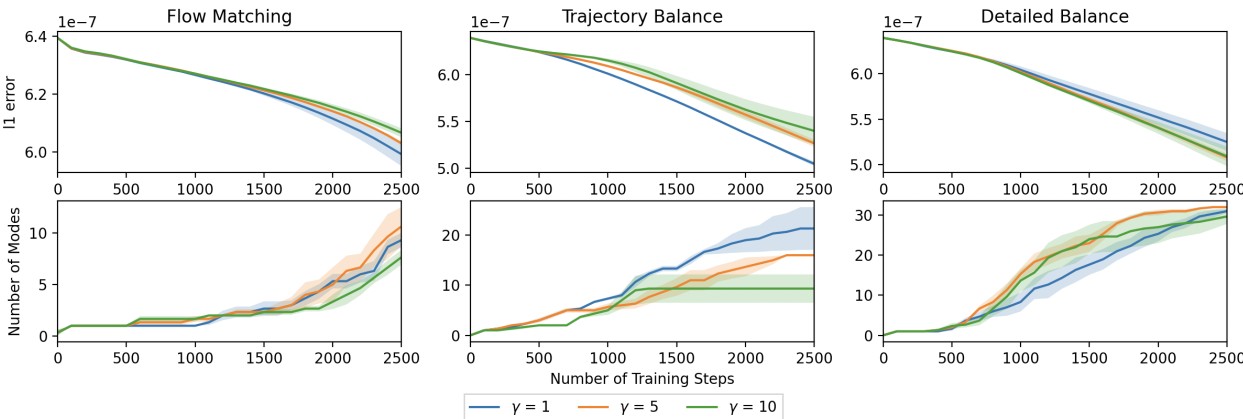

**Figure 18:** Experimental results for the hypergrid task for EGFN among different values of $\gamma$ across the three training objectives. *Top:* the $\ell_1$ error between the learned distribution density and the true target density. *Bottom:* the number of discovered modes across the training process. Here, $H = 20, D = 5$. We observe that lower $\gamma$ leads to better results for FM and TB, while higher $\gamma$ leads to better results for DB.

We now turn our attention to the mutation. To observe the effect of the mutation strength $\gamma$, we run the hypergrid experiment for the three training objectives using EGFN for $\gamma \in \{1, 5, 10\}$. The hypergrid configurations follow the the same configurations as before. We plot the mode discovery and $\ell_1$ error between the learned distribution density and the true target density over 2500 training steps in Figure 18. While the results indicate that having a higher $\gamma$ leads to better result for DB, the improvement is not extraordinary. Besides, in our work, we experience training instability for higher $\gamma$. Thus, we restrict $\gamma$ to be 1 throughout in our work.

> **Takeaway:** Higher mutation strength generally leads to diminishing returns.

## F.5 Priority percentile

PRB is an important component of our EGFN setup. Therefore, an important question is how the definition of priority samples stored in the replay buffer affects the results. To answer this question, we run the hypergrid experiment in the hardest setting while changing the priority percentile $\in \{50, 70, 90\}$ controlling the sampling split, i.e., while sampling offline trajectories from the PRB, 50% of them come from the priority samples and the rest come from the non-priority samples. Figure 19 shows the results. It indicates that having high-reward priority samples stored in the replay buffer benefits the training in terms of mode discovery, but there is little improvement in distribution fitting.

> **Takeaway:** Higher priority percentile improves mode-discovery of TB.

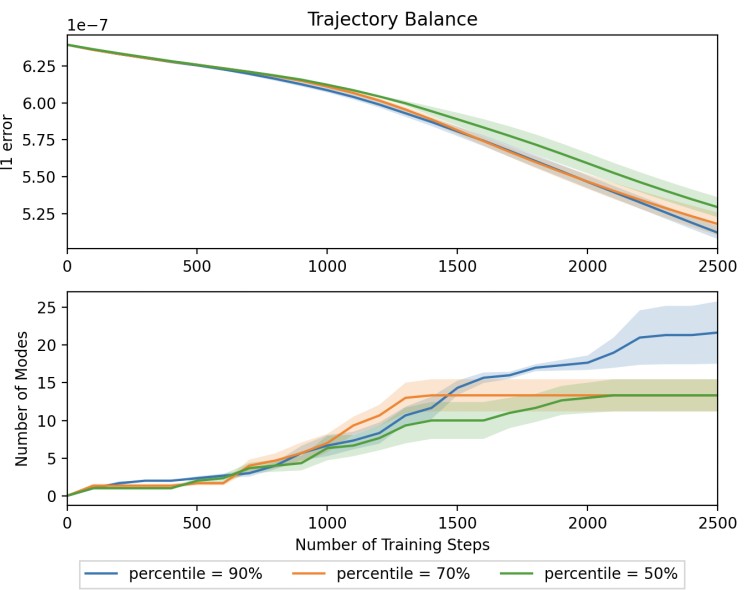

**Figure 19:** Experimental results for the hypergrid task for EGFN among different percentiles of priority samples for the TB objective. Throughout the experiment, we keep a 50-50 split between the priority and non-priority samples. *Top:* the $\ell_1$ error between the learned distribution density and the true target density. *Bottom:* the number of discovered modes across the training process. Here, $H = 20, D = 5$. We observe that higher percentile priority samples lead to better results for TB.

## F.6 Priority split

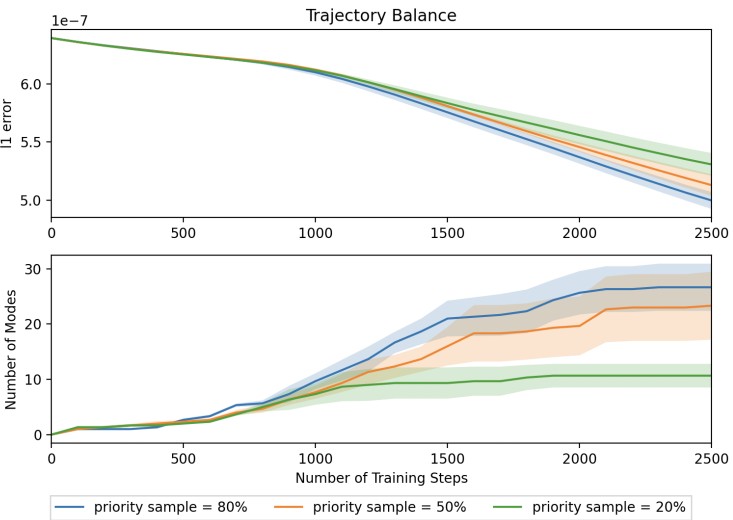

**Figure 20:** Experimental results for the hypergrid task for EGFN among different splits of priority sampling for the TB objective. Throughout the experiment, we used 90 percentile samples as priority samples. *Top:* the $\ell_1$ error between the learned distribution density and the true target density. *Bottom:* the number of discovered modes across the training process. Here, $H = 20, D = 5$. We observe that having a high priority-non-priority split leads to better results for TB.

Finally, a decision just as important as the priority percentile is the sampling ratio between the priority and non-priority offline trajectories. This time, we keep the priority percentile fixed at 90%, while changing

the priority to non-priority ratio splits between $\{20, 50, 80\}$. We run an experiment similar to the previous experiments with the aforementioned setting and report the results in Figure 20. The results clearly demonstrate that, despite their collection from a small subset of states, sampling more from the priority samples is beneficial for both mode discovery and distribution fitting.

> **Takeaway:** Higher priority-non-priority split improves mode-discovery and distribution-fitting of TB.

## G    Baselines

**MARS.**   MArkov moleculaR Sampling method (MARS) is a multi-objective molecular design method that define a Markov chain over the explicit molecular graph space and design a kernel to navigate high probable candidates with acceptance-rejection sampling.

**GAFN.**   Generative Augmented Flow Networks (GAFN) is a GFlowNets variant that aims to address the exploration challenge of GFlowNets by enabling intermediate rewards in GFlowNets and thus intrinsic rewards.

**SAC.**   Soft Actor-Critic (SAC) is an off-policy RL algorithm that maximizes a trade-off between expected reward and entropy, encouraging diverse and stable exploration.

**PPO.**   Proximal Policy Optimization (PPO) is a policy-gradient method that simplifies training by constraining policy updates, ensuring stable and reliable improvement during optimization.

**IQL.**   Implicit Q-Learning (IQL) is an offline reinforcement learning approach that effectively extracts value-based policies by decoupling value estimation from policy improvement, achieving stable learning from static datasets.

## H    Additional analysis

**Long time horzion results with reward evalutations.**   To understand the sample complexity of EGFN in comparison to GFlowNets, we plot the results for long time horizon task against the number of $R(x)$ calls for the default hyperparameters in this paper. We see that EGFN captures the $R(x)$ distribution better than GFlowNets with comparable trajectory evaluations.

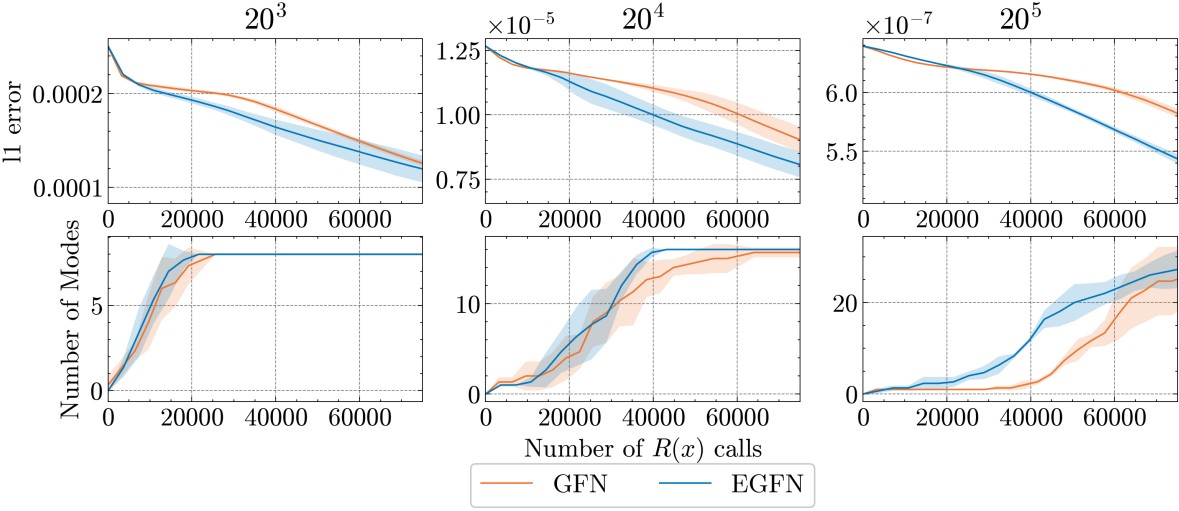

**Figure 21:** Experimental results comparison for the hypergrid task between EGFN and GFlowNets across increasing dimensions for 2500 training steps. The results shows that for similar amount of evaluation calls, EGFN performs better than GFlowNets with increasing difficulty.

**Population performances against the star agent.** To understand the contribution of the population during the training of the star agent, we plot the performances of the population and the star agent of EGFN while training in an increasingly complex environment. For fair comparison, we choose mean fitness for the population ($K = 4$) and top 10% reward for the star agent as a performance metric, as it is trained on a GFlowNets loss. The results in Figure 22 show that the population are more resistant to the increasing difficulty, which ultimately drives the PRB samples, improving GFlowNets training.

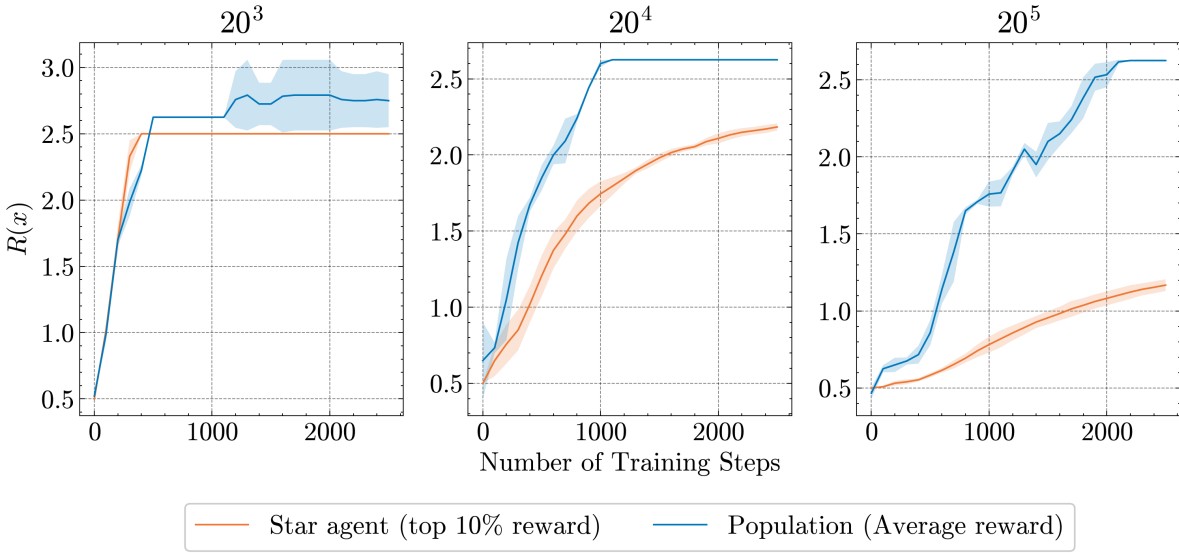

**Figure 22:** Comparing the star agent and population on increasing dimensions for 2500 training steps. For star agent, we report the top 10% reward, and for population, we report the mean reward.

# I  Additional algorithms

## I.1  Crossover

---
**Algorithm 3** Crossover step

---
**Data:** Agent weights $\theta_{P_{F_1}}$, $\theta_{P_{F_2}}$
**Result:** Crossover between agent weights
**Procedure** CROSSOVER($\theta_{P_{F_1}}$, $\theta_{P_{F_2}}$)

    **for** *weights $w_1$, $w_2$ in $\theta_{P_{F_1}}$, $\theta_{P_{F_2}}$* **do**

        N = NUM_ROW($w_1$)

        num_mutations $\sim \mathcal{U}(N)$

        **for** *count = 1 to num_mutations* **do**

            index $\sim \mathcal{U}(N)$

            **if** $r() < 0.5$ **then**

                $w_1[index] = w_2[index]$

            **else**

                $w_2[index] = w_1[index]$

---

