# OpenReview forum: "Evolution guided generative flow networks"
_TMLR — Accepted by TMLR_

### Review · Reviewer_VDKA · 2025-01-23

**Summary Of Contributions:**

The paper trains investigates training a population of GFlowNets with the evolutionary algorithm in Alg 1, which maintains a population of networks and selects the elite ones to mutate, interspersed with sampling trajectories and evaluating the performance. This is experimentally compared on a grid MDP (Fig 3/4/5), antibody sequence optimization (Fig 6/Table1), sEH binder generation (Fig 7), and hu4D5 CDR H3 mutation generation.

**Audience:**

Yes

**Broader Impact Concerns:**

No concerns

**Claims And Evidence:**

Yes

**Requested Changes:**

None

**Strengths And Weaknesses:**

The paper is a well-documented exploration on evolutionary methods for GFlowNets. I find all of the claims to be backed up and the experimental evidence suitable and interesting enough for the TMLR audience.  On the novelty, it is a standard evolutionary method, but I am not aware of other works investigating it in this way. And, the extensive experimental analysis and ablations throughout the appendix will be useful to practitioners.

One question I have: is it fair to compare the number of training steps between all of the methods in the experiments? My minor concern is that the methods use significantly different information for each update, e.g., spanning GFN, EGFN, PPO, SAC, and others, and it seems like some of them may be more computationally expensive than others.

---

> ### Author Response · Authors · 2025-04-30
> **Q1 and Q2**
>
> Q1. Is it fair to compare the number of training steps between all of the methods in the experiments?
>
> A1. To ensure fairness from a sample efficiency perspective, we kept the total training samples in step the same. In fact, in terms of reward calls, they are also roughly the same, e.g., EGFN uses 36 calls. Other baselines use 32, except for PPO which doubles the reward calls since we double the amount of on-policy samples due to not having off-policy samples.
>
> For more rigorous analysis, we have included preliminary results with number of trajectory evaluations on the X axis comparing GFlowNets and EGFN on __Figure 21__ in __Appendix H__ with the same hyper parameters of our experiments. We still see significant improvement of EGFN, compared of GFlowNets.
>
> Q2. Methods use significantly different information for each update, e.g., spanning GFN, EGFN, PPO, SAC, and others, and it seems like some of them may be more computationally expensive than others.
>
> A2. Of course, in terms of wall-clock efficiency, our algorithm is _slower_, which we indicate in our work. However, in terms of learning a distribution with many peaks, it is likely useful to _explore_ more taking time, than missing multiple peaks.
>
> For example, authors in [1] show that, "EA ... solves the (long-horizon) task after 22000 episodes. DDPG on the other hand fails to solve the task entirely."
>
> This phenomenon of standard GFlowNets (or the other baselines) stuck around the same mode can be explained with divergent search techniques too, as the authors in [2] mention "combining extinction events with divergent search increases
> evolvability, while combining them with convergent search oﬀers no similar benefit."
>
> 1. Khadka, Shauharda, and Kagan Tumer. "Evolution-guided policy gradient in reinforcement learning." Advances in Neural Information Processing Systems 31 (2018).
> 2. Lehman, Joel, and Risto Miikkulainen. "Enhancing divergent search through extinction events." Proceedings of the 2015 Annual Conference on Genetic and Evolutionary Computation. 2015.

---

### Review · Reviewer_9dFT · 2025-02-28

**Summary Of Contributions:**

The paper aims to improve the effectiveness of GFlowNets in long horizon and sparse reward settings. The key contribution is to use a separate population of GFlowNet agents to provide off-policy samples in addition to the on-policy samples. This population is refined by evolutionary algorithms (including selection, crossover, mutation). Experiments on synthetic and real-world benchmarks show that the proposed method can discover more modes with fewer training steps and can better match the target density.

**Audience:**

Yes

**Claims And Evidence:**

No

**Requested Changes:**

- Clarify what samples are used to compute the L1 error and the number of modes discovered
- Preferably increase the reward sparsity by shrinking the rewarding region
- Improve presentation, fix typos and inconsistencies

**Strengths And Weaknesses:**

Strengths
- The paper conducts a wide range of experiments, and shows strong results compared to quite a few baselines including SAC, original GFlowNets and more recent variants. The paper also has quite extensive ablation study, including effects of prioritized replay, mutation, crossover, and using random ensembles instead of evolution guided population for off-policy sampling.

Weaknesses
- The evaluation protocol is not that clear to me. For example, in Fig 3 and 4, what samples are used to compute the L1 error and the number of modes discovered? Do you use samples from the star agent only or do you also use samples from the replay buffer / EA population? If the latter, the comparison might be unfair.
- Regarding the reward sparsity experiment, I think a more natural way to make the reward more sparse is to make the region of R1 and R2 smaller, i.e., making the modes less accessible by a random agent.
- Presentation could be improved.
    - Sec 3.1 could provide more details of crossover, i.e., how is crossover implemented between two agents? Is it simply swapping all the weights uniformly at random?
    - In Fig 3, the colors are not easily distinguishable. GFN and IQL seem to have the same color. MARS and SAC also have quite similar colors. In Fig 5, GFN and EGFN have the same color.
    - It would be better to put Eq 6 in the main text rather than appendix.
    - It looks like there is a typo in the ablation study setup (Sec 4.1, Setup). Should H=16 be H=20 instead?
    - The $R_0$ in Fig 9 caption is different from what is stated in the main text (first line below Table 2).
- In Fig 9, why can the trajectory length be less than 0?

---

> ### Author Response · Authors · 2025-04-30
> **Q1-Q4**
>
> Q1. The evaluation protocol is not that clear to me. For example, in Fig 3 and 4, what samples are used to compute the L1 error and the number of modes discovered? Do you use samples from the star agent only or do you also use samples from the replay buffer / EA population? If the latter, the comparison might be unfair.
>
> A1. We evaluate the star agent only by keeping state counts visited by the star agent. This is consistent with previous works. For example, we follow [1] codebase for evaluation for the initial experiments while for others, we follow [2] codebase.
>
> Q2. Regarding the reward sparsity experiment, I think a more natural way to make the reward more sparse is to make the region of R1 and R2 smaller, i.e., making the modes less accessible by a random agent.
>
> A2. This is similar to what we do. Since we are working with unnormalized reward distribution, a lower $R_0$ means the _normalized_ region around the peak will steeper, meaning the peaks will be shaper. This has the same effect of making the modes less accessible by a random agent.
>
> Besides, this approach of achieving reward sparsity is inspired from previous GFlowNets works [1, 3].
>
> Q3. Sec 3.1 could provide more details of crossover, i.e., how is crossover implemented between two agents? Is it simply swapping all the weights uniformly at random?
>
> A3. Crossover simply swaps randomly chosen weights. We added one line in __Section 3.1__ to reflect this. Besides, we also added the detailed algorithm in __Appendix I__.
>
> Q4. In Fig 3, the colors are not easily distinguishable. GFN and IQL seem to have the same color. MARS and SAC also have quite similar colors. In Fig 5, GFN and EGFN have the same color.
>
> A4. We thank the reviewer for pointing this out. To address the concern, we have re-done the __Figure 3__ and __5__ with different color cycles.
>
>
> 1. Zhang, Dinghuai, et al. "Distributional GFlowNets with Quantile Flows." Transactions on Machine Learning Research.
> 2. Shen, Max W., et al. "Towards understanding and improving gflownet training." International conference on machine learning. PMLR, 2023.
> 3. Malkin, Nikolay, et al. "Trajectory balance: Improved credit assignment in gflownets." Advances in Neural Information Processing Systems 35 (2022): 5955-5967.

---

> ### Author Response · Authors · 2025-04-30
> **Q5-Q8**
>
> Q5. It would be better to put Eq 6 in the main text rather than appendix.
>
> A5. We moved the __Equation 6__ in the main text in __Section 4.1__.
>
> Q6. It looks like there is a typo in the ablation study setup (Sec 4.1, Setup). Should H=16 be H=20 instead?
>
> A6. Indeed it was a typo, we have edited our manuscript accordingly. We thank the reviewer for their careful observation.
>
> Q7. The $R_0$  in Fig 9 caption is different from what is stated in the main text (first line below Table 2).
>
> A7. We again thank the reviewer for their careful reading. It is a typo. We experimented with $R_0 = $ $10^{-2}$ and $10^{-5}$. We have edited our manuscript to reflect the typo-correction.
>
> Q8. In Fig 9, why can the trajectory length be less than 0?
>
> A8. The trajectory lengths are not less than 0. Rather, due to KDE plotting, the kernel density estimates are showing smooth lines that goes under 0.

---

### Review · Reviewer_mpvD · 2025-04-20

**Summary Of Contributions:**

This paper introduces Evolution-guided Generative Flow Networks (EGFN), an augmentation framework that enhances GFlowNets with Evolutionary Algorithms (EA). The approach leverages gradient-free EA to select a set of trajectories from multiple agents, which are then used to train a central ‘star-agent’ through gradient-based optimization. The authors validate the effectiveness of their method on both synthetic and real-world datasets, showing that it outperforms baseline models, as the cost of increased inference time.

**Audience:**

Yes

**Broader Impact Concerns:**

There are no broader impact concerns.

**Claims And Evidence:**

Yes

**Requested Changes:**

My primary concern with this submission is the connection between the proposed method and the experiments presented. I believe the authors should clarify this relationship more clearly and include additional supporting experiments to better emphasize their contribution.
Please refer to the questions listed above under the weaknesses section.

**Strengths And Weaknesses:**

**Strengths**
1. The proposed approach combines both GFlowNets and AE into a single training framework to get the best of both worlds.
2. Results suggest the proposed approach is superior to the evaluated baselines (results are reported for both real and synthetic data).

**Weaknesses**
1. The paper is somewhat difficult to follow. Since I am not a GFlowNets expert, a more comprehensive and motivational background section would greatly help in understanding the core ideas.
2. While the authors report results across multiple datasets, the paper lacks a thorough ablation or analytical component. For instance, if my understanding is correct, a central contribution of the work is the integration of Evolutionary Algorithms (EA). In that case, it would be helpful to analyze the impact of various EA components:
* How does the population size influence model performance?
* What are the specific fitness, mutation, and crossover functions used, and how do they affect the outcomes?
* What is the effect of selecting the different $\epsilon$% elite agents?
* What proportion of samples are drawn from $P^*_F$, and what is their influence on model performance?

Currently, Algorithm 1 offers the most clarity in explaining the method. I encourage the authors to similarly clarify these aspects in the main text.

3. The abstract states: “One big challenge of GFlowNets is training them effectively when dealing with long time horizons and sparse rewards.” However, it is unclear how the proposed method addresses this challenge. A more explicit explanation would strengthen the contribution.
4. Can the proposed EA-based method be extended to train models with other reward-driven objectives, such as PPO?

---

> ### Author Response · Authors · 2025-04-29
> **Q1-Q3**
>
> Q1. The paper is somewhat difficult to follow. Since I am not a GFlowNets expert, a more comprehensive and motivational background section would greatly help in understanding the core ideas.
>
> A1. We thank the reviewer for bringing this point. The key motivation of GFlowNets can be summarized by attempting to capture the normalized reward distribution unlike traditional RL where we are only focused on the reward peaks. The key motivation for our work is scaling up GFlowNets for higher dimensions by utilizing EA's quality-diversity searching capabilities.
>
> Q2. How does the population size influence model performance?
>
> A2. We analyze population size's relation to the performances of different GFlowNets criterions in Section F1. Higher population provides higher diversity; this improves FM and DB's performance. We have also added "Takeaway" block in the manuscript for easy access for practitioners.
>
> Q3. What are the specific fitness, mutation, and crossover functions used, and how do they affect the outcomes?
>
> A3. We thank the reviewer for this question. As we are using EA for Neural networks used by the agents' policy, we generally follow standard practices in EA.
>
> For fitness evaluation of each agent, we use the mean reward received by $\mathcal{E}$ terminal states sampled from the agent. We draw the reviewer's attention to the __Algorithm 2__ of __Section C__, where we detail the Fitness evaluation process. Intuitively, we are simply improving the agent's mode-seeking behavior by approximating its fitness through limited samples, hoping to achieve better quality-diversity. Indeed, this is important for the _star-agent_, as [1] mentions "...diversity-promoting techniques often improve the quality of the solutions produced and the number of different types of solutions explored"
>
> For mutations, we use a simple mutation strategy for weights. For a mutation probability, we simply add weight matrix sampled from a zero-mean normal distribution $\mathcal{N}(0, \gamma)$.
>
> To promote diversity, we use a crossover strategy inspired from [2]. For a number of crossover occurrences, we flip the weights of randomly chosen neurons of two agents given some chosen probability. For a clear understanding of this approach, we have also added the detailed crossover approach in the __Algorithm I1__ of __Section I__.
>
> 1. Jean-Baptiste Mouret and Jeff Clune. "Illuminating search spaces by mapping elites."
> 2. Khadka, Shauharda, and Kagan Tumer. "Evolution-guided policy gradient in reinforcement learning." Advances in Neural Information Processing Systems 31 (2018).

---

> ### Author Response · Authors · 2025-04-30
> **Q4-Q7**
>
> Q4. What is the effect of selecting the different $\epsilon$% elite agents?
>
> A4. We analyze the effect of selecting the different $\epsilon$% elite agents in __Section F2__. We trial with elite population percentage of 0.2, 0.4, and 0.6 for flow matching, trajectory balance, and detailed balance. We report the result in __Figure 16__ where we show that having reasonably small elite population percentage helps FM and TB through lower mode-seeking behavior.
>
> Indeed, this is also consistent with previous literature. For example, [1] keeps "Proportion of best agents" to 0.1.
>
> Q5. What proportion of samples are drawn from $P_F^*$, and what is their influence on model performance?
>
> A5. We use ~40% samples from $P_F^*$. The samples ensure stability, as a complete off-policy training samples can hamper the star agent's learning progress. On the other hand, a fully on-policy training turns our algorithm to the standard GFlowNets training.
>
> Q6. The abstract states: “One big challenge of GFlowNets is training them effectively when dealing with long time horizons and sparse rewards.” However, it is unclear how the proposed method addresses this challenge. A more explicit explanation would strengthen the contribution.
>
> A6. EA provides high-reward diverse samples for PRB to sample from, allowing GFlowNets agent to learn from a diverse quality samples, capturing the reward distribution better in case of  long time horizons and sparse rewards.
>
> Indeed, credit assignment in long trajectories and sparse rewards is difficult with the current advances, as in [2] the authors conclude, "TB trades off the advantage of immediately providing credit to early states with the disadvantage of relying on sampling of long trajectories...".
>
> This problem is evident in other methods too, as the authors in [3] discuss, "... methods in RL use bootstrapping to address this issue but often struggle when the time horizons are long and the reward is sparse"
>
> This being the case, we address it in the difficult cases and utilize current credit assignment methods for our method. Our evaluations in __Figure 9__ show that the mentioned cases make the training trajectory lengths skewed, and utilizing EA helps it to be more balanced by improving diversity.
>
> In the __Figure 22__ of the __Appendix H__, we compare the rewards between the population and the star agent, showing that the population achieves high mean reward quicker than the star agent. It can also be seen that with increasing difficulty, while the star agent's top 10% reward starts to fall, the EA population stays more consistent with finding high-reward samples for PRB, given that it is a black-box optimization method.
>
> Q7. Can the proposed EA-based method be extended to train models with other reward-driven objectives, such as PPO?
>
> A7. Yes! In fact, a very recent work from March'25 shows that it could work for PPO [4]. Here, authors mention "EPO maintains a population of policies alongside a central master policy, all sharing the same network weights (analogous to gene-expression rules)", which is similar to our work.
>
> 1. PIERROT, Thomas, and Arthur Flajolet. "Evolving Populations of Diverse RL Agents with MAP-Elites." The Eleventh International Conference on Learning Representations.
> 2. Malkin, Nikolay, et al. "Trajectory balance: Improved credit assignment in gflownets." Advances in Neural Information Processing Systems 35 (2022): 5955-5967.
> 3. Khadka, Shauharda, and Kagan Tumer. "Evolution-guided policy gradient in reinforcement learning." Advances in Neural Information Processing Systems 31 (2018).
> 4. Wang, Jianren, et al. "Evolutionary Policy Optimization." arXiv preprint arXiv:2503.19037 (2025).

---

### Author Response · Authors · 2025-05-11
**Summary of rebuttal**

Dear Action Editor,

We appreciate the reviewers' thoughtful evaluations of our paper and have carefully addressed their concerns. Below, we provide a unified discussion of the main issues raised and our responses.
______

Main Concerns and Our Responses:

- Clarification of method details:
  - Concern: Some reviewers asked for more details on key parts of EA algorithm used in our work, experiment details, etc.
  - Response: We clarified the details on EA. We added detailed crossover approach in the Algorithm I1 of Section I. We also clarified the mutation strategy, describing that we use a simple mutation strategy for weights. For evaluation protocol, we confirmed that we evaluate the star agent only by keeping state counts visited by the star agent, taking inspiration from previous works.

- Takeaway from ablation experiments on EA steps
  - Concern: Reviewer mpvD asked for analysis on the impact of various EA components.
  - Response:  We analyze population size's relation to the performances of different GFlowNets criterions in __Section F1__. We analyze the effect of selecting the different % elite agents in __Section F2__. We have also added _takeaway_ block for all our ablation experiments, providing easy understanding for practitioners.

- Fairness of experiments
  - Concern: Questions were raised about how the proposed method addresses the challenge of training in long time horizons and sparse rewards. Reviewer 9dFT asked about what samples were used to compute the L1 error and the number of modes discovered. Reviewer VDKA asked whether it is fair to compare the number of training steps between all of the methods in the experiment.
  - Response: We clarified that we address the training problem in the difficult cases and utilize current credit assignment methods for our method, pointing to __Figure 9__ that shows that the mentioned cases make the training trajectory lengths skewed, and utilizing EA helps it to be more balanced by improving diversity. We also incorporate more experiment in the __Figure 22__ of the __Appendix H__, where we compare the rewards between the population and the star agent, showing that the population achieves high mean reward quicker than the star agent. We confirm that we evaluate the star agent only by keeping state counts visited by the star agent, taking inspiration from previous works.
  - Finally, we confirm that to ensure fairness from a sample efficiency perspective, we kept the total training samples in step the same. Additionally, we include preliminary results with number of trajectory evaluations on the X axis comparing GFlowNets and EGFN on __Figure 21__ in __Appendix H__, still showing significant improvement of EGFN, compared of GFlowNets.

- Presentations
  - Concern: Reviewer 9dFT pointed out several presentation problems.
  - Response: We fixed the typos, made the plots more easily distinguishable with color, and clarified several parts of manuscript with better writing.

We believe that our revisions have effectively addressed the reviewers' concerns and have strengthened the paper's clarity and contribution to the field.

We kindly ask the action editor to consider our responses and the improvements made.

Thank you for your consideration!

---

> ### Comment · Action_Editor_uyhh · 2025-06-12
> **Comment on rebuttal**
>
> Dear authors,
>
> Thank you for these responses. Having looked at the submission, the reviews, and your responses, there is one additional point I would like to raise:
>
> I note that both reviewers 9dFT and mpvD had quite a few questions about details of implementation and evaluation. Many of your responses point to additional details to sections of the appendix, but as far as I can see only sections C, D.1 and D.2 are referenced in the main text of your submission. Could you please make sure you reference remaining sections in the appendix in the relevant context in the main text?
>
> With thanks,
> The AC

---

> > ### Author Response · Authors · 2025-06-12
> > **Updated manuscript**
> >
> > Dear AE,
> >
> > We have provided an update to our manuscript to reflect the changes you requested. We have referenced the sections from the appendix wherever necessary. For example, we have revised the section 5 to provide context to the experiments presented in the appendix H.
> >
> > Please let us know if you have any other recommendations or questions.
> >
> > Best,
> >
> > The authors

---

### Decision · Action_Editor_uyhh · 2025-06-12

**Recommendation:** Accept with minor revision

**Additional Comments:**

As noted in the response to the summary of the rebuttal, the AE would like to ask the authors to update the main text to ensure that relevant sections in the appendix are referenced where possible. Reviewers raised many questions about implementation and evaluation, which in many cases could be answered based on information in the appendix, but as far as the AE was able to ascertain, the main text often does not indicate when additional details have been relegated to the appendix.

**Audience:**

Yes

**Audience Explanation:**

An evaluation of the relevance of evolutionary algorithms in the training of GFNs is something that the AE would consider relevant to TMLR's audience.

**Claims And Evidence:**

Yes

**Claims Explanation:**

The paper trains investigates training a population of GFlowNets with the evolutionary algorithm. There are no particular claims pertaining to the novelty of evolutionary algorithm, the claimed contribution is its application to the training of GFlowNets. Reviewers raised questions about point of implementation and evaluation, which in many cases were addressed in (unreferenced) sections of the appendix, but expressed no substantial concerns about technical correctness and were overall considered the set of experiments relative extensive.